# Wind Turbine Fault Detection Through Autoencoder-Based Neural Network and FMSA

**DOI:** 10.3390/s25144499

**Published:** 2025-07-19

**Authors:** Welker Facchini Nogueira, Arthur Henrique de Andrade Melani, Gilberto Francisco Martha de Souza

**Affiliations:** Department of Mechatronics and Mechanical Systems Engineering, Polytechnic School, University of Sao Paulo, Sao Paulo 05508-010, SP, Brazil; welkerfacchini@usp.br (W.F.N.); melani@usp.br (A.H.d.A.M.)

**Keywords:** wind turbine, fault detection, predictive maintenance, FMSA, autoencoders

## Abstract

Amid the global shift toward clean energy, wind power has emerged as a critical pillar of the modern energy matrix. To improve the reliability and maintainability of wind farms, this work proposes a novel hybrid fault detection approach that combines expert-driven diagnostic knowledge with data-driven modeling. The framework integrates autoencoder-based neural networks with Failure Mode and Symptoms Analysis, leveraging the strengths of both methodologies to enhance anomaly detection, feature selection, and fault localization. The methodology comprises five main stages: (i) the identification of failure modes and their observable symptoms using FMSA, (ii) the acquisition and preprocessing of SCADA monitoring data, (iii) the development of dedicated autoencoder models trained exclusively on healthy operational data, (iv) the implementation of an anomaly detection strategy based on the reconstruction error and a persistence-based rule to reduce false positives, and (v) evaluation using performance metrics. The approach adopts a fault-specific modeling strategy, in which each turbine and failure mode is associated with a customized autoencoder. The methodology was first validated using OpenFAST 3.5 simulated data with induced faults comprising normal conditions and a 1% mass imbalance fault on a blade, enabling the verification of its effectiveness under controlled conditions. Subsequently, the methodology was applied to a real-world SCADA data case study from wind turbines operated by EDP, employing historical operational data from turbines, including thermal measurements and operational variables such as wind speed and generated power. The proposed system achieved 99% classification accuracy on simulated data detect anomalies up to 60 days before reported failures in real operational conditions, successfully identifying degradations in components such as the transformer, gearbox, generator, and hydraulic group. The integration of FMSA improves feature selection and fault localization, enhancing both the interpretability and precision of the detection system. This hybrid approach demonstrates the potential to support predictive maintenance in complex industrial environments.

## 1. Introduction

In recent years, the growing concern over environmental degradation has driven the increasing prominence of sustainability topics in modern society [1]. The dependence on fossil fuels for electricity generation, characterized by environmental harm and resource finitude, underscores the urgency of transitioning to renewable sources. Among these, wind energy has emerged as a key pillar for building a more sustainable energy future. Technological development is, thus, essential to enhance the efficiency of renewable sources and facilitate their broader integration.

The global wind energy sector has experienced continuous growth, with the Global Wind Energy Council reporting a cumulative installed capacity of 77.6 GW in 2022 and projections exceeding 100 GW by 2024 [2]. Despite this significant growth, ensuring operational reliability and effective maintenance strategies remains challenging, particularly due to the remote and often inaccessible locations of wind farms, complicating routine inspections and interventions.

Effective asset maintenance is essential to guarantee the continuous and efficient operation of wind turbines. Recent advances in artificial intelligence (AI) and data-driven technologies have substantially improved predictive maintenance, especially in fault detection and diagnosis tasks. Data-driven approaches leverage operational data from supervisory control and data acquisition (SCADA) and condition monitoring systems (CMSs) to monitor asset health in real time. Among these methodologies, deep learning techniques, particularly autoencoder-based neural networks, have demonstrated promising results in anomaly detection applications. Autoencoders effectively model complex patterns in high-dimensional data without relying extensively on labeled datasets.

Previous studies have demonstrated the effectiveness of autoencoders in various applications. For instance, in the study by Cui and Tjernberg [3], autoencoders and recurrent units were used to model the normal behavior of power transformers, enabling the early detection of operational risks before sensor faults occurred.

Similarly, Radaideh et al. [4] explored the use of recurrent and convolutional LSTMs autoencoders for anomaly detection in time series signals from power electronics. Additionally, Qi et al. [5] proposed a fault detection method for arc faults using wavelet transform and LSTM autoencoders for photovoltaic energy systems.

Huynh et al. [6] developed an unsupervised framework using multilayer autoencoders for anomaly detection in power grids. Rosa et al. [7] also employed deep neural networks based on autoencoders for fault detection in hydroelectric plants. Their work was later expanded to include failure prognosis in the proposed framework, demonstrating that the remaining useful life of components can be predicted semi-supervised using autoencoders.

Tang et al. [2] propose a hybrid approach for fault diagnosis in wind turbine generators, combining a stacking algorithm with an adaptive threshold mechanism. The model is structured into two layers: In the first, some base classifiers are employed; in the second layer, a logistic regression-based meta-classifier integrates these outputs, forming the decision-making structure. The authors also introduce an adaptive threshold calculated from the mean and standard deviation of the estimated fault probability during the machine’s healthy condition. The methodology was initially validated on simulated data from an experimental test bench using a three-phase asynchronous generator subjected to induced faults. Subsequently, the approach was tested on a real-world dataset provided by EDP, focusing on Turbine T07, which presented a recorded generator bearing failure.

Bindingsb et al. [8] propose a methodology based on interpretable machine learning for fault detection in wind turbine generator bearings. The technique involves predicting bearing temperature under normal operating conditions. To ensure model interpretability, the authors employ the Shapley Additive Explanations tool, enabling the identification of each input variable’s contribution to the model’s predictions and enhancing confidence in the obtained results. Several algorithms, including linear regression, random forest, support vector regression, and XGBoost, were tested and compared, with XGBoost ultimately selected as the most effective, based on metrics such as RMSE, MAE, and execution time. The methodology was applied to dataset provided by EDP, with the case study focusing on Turbine T07. The results demonstrate that the model is capable of detecting failures in advance.

Du et al. [9] present a denoising autoencoder designed to detect anomalies in wind turbines. The DAE is trained on historical normal data to enhance robustness against noise, and the reconstruction error is evaluated using an exponentially weighted moving average control chart to reduce false positives. The model’s performance is assessed through sparse fault estimation, which identifies the key variables contributing to the detected anomalies.

Chokr et al. [10] propose an approach for anomaly detection in wind farms using a bidirectional long short-term memory autoencoder, designed for high-dimensional multivariate time series data. The model was trained and tested with real SCADA data from five wind turbines, demonstrating its ability to detect anomalies based on normal data. The study highlights the importance of capturing temporal dependencies and the challenges in selecting detection thresholds, contributing to improved safety and reliability in renewable energy systems.

Miele, Bonacina, and Corsini [11] propose an anomaly detection approach for wind turbines using an autoencoder applied to multivariate time series derived from SCADA systems. The proposed architecture consists of an encoder and decoder based on convolutional neural networks. Input data are organized into sliding windows. The model is trained to reconstruct the machine’s normal behavior. The detection methodology comprises two main stages: (i) the definition of a global Mahalanobis indicator, which assesses global multivariate deviations based on the reconstruction error distribution during normal operation and (ii) the calculation of local residual indicators, which quantify reconstruction errors for each monitored features. The model was validated using a publicly available real SCADA dataset provided by EDP. MTGCAE was benchmarked against two reference architectures from the literature, LSTM-AE and CNN–LSTM, demonstrating superior performance across all evaluation metrics.

Nogueira et al. [12] propose a fault detection methodology for wind turbines based on autoencoder neural networks applied to SCADA time-series data. The approach consists of a structured six-step procedure. The autoencoder was implemented using a multi-layer perceptron architecture and trained with only healthy operational data. Anomalies were identified through the reconstruction error, quantified by the root mean squared error, and a static detection threshold was defined as the mean RMSE plus three standard deviations. The methodology was validated using a real-world SCADA dataset from the EDP Open Data. However, only one failure mode was analyzed: overheating of the high-voltage transformer of Turbine T01, caused by fan degradation, which led to a total turbine shutdown. Although Failure Mode and Symptoms Analysis (FMSA) was not performed, the authors recognize its potential for future integration to improve feature selection and enhance fault localization. The model successfully detected a deviation in the reconstruction error before the failure recorded in the log, indicating its anomaly detection capability in wind turbine monitoring systems.

Despite the strong performance of autoencoder-based models in anomaly detection, their practical applicability heavily depends on the interpretability of results and the careful selection of input data, highlighting the need for integrating structured methods to support these tasks. The primary objective of this study is to develop and validate a novel fault detection framework for wind turbines by integrating autoencoder-based neural networks with Failure Mode and Symptoms Analysis (FMSA). This integration addresses one of the main limitations of data-driven approaches: the lack of physical meaning behind the input variables and the resulting anomalies. FMSA plays a central role in this framework by guiding the feature selection process based on engineering knowledge and historical failure evidence. Unlike purely statistical or correlation-based methods, FMSA enables the identification of measurable symptoms that are causally linked to specific failure mechanisms. Moreover, it allows the construction of interpretable models even in the absence of labeled fault data, supporting the early deployment of detection tools. In real-world scenarios, where faults may not have occurred yet in the monitored turbine, FMSA provides a proactive and structured way to isolate the most relevant input signals, enhancing the observability of degradation patterns and improving the reliability of the resulting anomaly detection system. As shown in Table 1, this approach distinguishes itself from recent studies by systematically incorporating expert knowledge through FMSA, which is rarely addressed explicitly in the existing literature.

To demonstrate the applicability and effectiveness of the proposed methodology, the developed framework was applied to two complementary case studies. The first involved simulated data generated via OpenFAST, allowing the validation of the model under controlled conditions with a known fault scenario. The second study relied on real operational data provided by the EDP Open Data platform, encompassing historical SCADA records and documented failures from actual wind turbines. This dual approach enables a comprehensive assessment of the method’s robustness and generalizability.

The novelty of this research resides in its systematic integration of autoencoders with FMSA, an approach scarcely explored in existing literature. While previous studies primarily rely on either deep learning architectures or interpretability-driven methods separately, the present study uniquely combines these elements to enhance the accuracy, interpretability, and reliability of fault detection. By explicitly linking observed anomalies identified by autoencoders with specific fault symptoms and degradation patterns through FMSA, this work provides a comprehensive and actionable diagnostic tool, capable of supporting informed and proactive maintenance decisions in wind energy operations.

## 2. Theoretical Background

This chapter introduces the theoretical foundations of the proposed methodology. First, the Failure Mode and Symptoms Analysis (FMSA) method is briefly described, followed by an explanation of autoencoder neural networks. These two techniques form the analytical basis for the fault detection approach developed in this study.

### 2.1. Failure Modes and Symptoms Analysis

FMSA is a structured approach used to systematically identify potential failures and their observable symptoms in operational data. According to ISO 13379 [13], FMSA enables the mapping of specific failure modes to measurable operational symptoms, supporting the selection of relevant features for fault detection models.

The FMSA process follows the main steps outlined below:Each component is analyzed based on historical records and technical insights to identify potential failure modes and the mechanisms that lead to their occurrence.For each identified failure mode, associated symptoms are mapped. These may range from physical signs, such as vibrations or abnormal noise, to anomalies observed in operational data.The severity of each failure, in terms of its impact on turbine operation and safety, is assessed. This evaluation supports the prioritization of critical failures and the identification of symptoms that require monitoring.

In contrast to traditional methodologies such as Failure Mode and Effects Analysis (FMEA) or Failure Mode, Effects, and Criticality Analysis (FMECA), FMSA specifically emphasizes symptom-based detection and monitoring strategies, assessing the effectiveness of different monitoring techniques in identifying failure symptoms rather than focusing solely on failure occurrences [14]. Gonzalez et al. (2023) [15] applied FMSA combined with FMECA to evaluate monitoring techniques in feed-drive systems, demonstrating the importance of selecting suitable detection strategies to maximize confidence and reduce unnecessary sensor deployment.

FMSA can also be employed to establish robust correlations between possible failure modes and specific signals monitored in industrial systems, such as those collected by SCADA systems in wind farms. By linking each identified failure mode with measurable symptoms, such as abnormal temperature fluctuations or excessive vibration, the analysis supports the technical validity and interpretability of the features used in anomaly detection models. Unlike traditional applications that emphasize prioritization and risk scoring, FMSA can serve as an analytical tool to support attribute selection. This aligns with recent literature [14,15] highlighting the benefits of symptom-based clustering in enhancing the sensitivity and specificity of predictive models, particularly those employing unsupervised techniques like autoencoders.

### 2.2. Autoencoder

The autoencoder is a specific type of neural network, known for its ability to reconstruct input data and is widely used in tasks such as compression, reconstruction, and anomaly detection [16]. It is an unsupervised representation learning algorithm in which the target values are set equal to the inputs. To prevent the autoencoder from learning a trivial identity mapping by simply copying the input data directly to the output without extracting any meaningful information, it is necessary to constrain the number of neurons in the hidden layers. This limitation forces the network to compress the data into a lower-dimensional representation, thereby preserving only the most relevant or informative aspects of the input. These aspects correspond to latent patterns, correlations, or structures present in the data, while noise and irrelevant variations tend to be discarded.

The autoencoder consists of two main components: the Encoder $f:R^{n}\to R^{d}$ and the Decoder $g:R^{d}\to R^{n}$. The Encoder maps the input data $x\in R^{n}$ to a compact latent representation $h\in R^{d}$, whereas the Decoder reconstructs the data from this compressed latent representation back to its original space $\bar{x}\approx x$. During training, the objective is to minimize a loss function that measures the discrepancy between the input data and their corresponding reconstructions at the output. This process forces the model to learn a compressed representation that retains the most relevant information from the original data. By minimizing this loss function using gradient-based optimization algorithms, the model parameters in both the encoder and the decoder are iteratively adjusted to improve the reconstruction capability of the network. For the backpropagation process to work properly, both components of the autoencoder must consist of continuous and differentiable functions [9].

Since the model proposed in this work is based on artificial neural networks, formal definitions are appropriate. According to [17], a feedforward neural network can be defined as A=(L,W,B), where $L=\{L_{1},L_{2},\ldots ,L_{r}\}$ represents the set of *r* layers, with each layer $L_{i}=\left\{x_{i1},x_{i2},\ldots ,x_{im_{i}}\right\}$ composed of $m_{i}$ neurons; $W=\{W^{\left(1\right)},W^{\left(2\right)},\ldots ,W^{\left(r-1\right)}\}$ is the set of weight matrices connecting each layer $L_{i}$ to the subsequent layer $L_{i+1}$; and $B=\{b^{\left(1\right)},b^{\left(2\right)},\ldots ,b^{\left(r-1\right)}\}$ is the set of bias vectors applied to each respective layer. The activation of the *j*-th neuron in layer $L_{i+1}$ is computed through the general feedforward propagation equation:
(1)$$x_{(i+1),j}=f^{\left(i\right)}\left(b_{j}^{\left(i\right)}+\sum_{k=1}^{m_{i}}x_{ik}w_{jk}^{\left(i\right)}\right)$$
where $f^{\left(i\right)}$ is the activation function of layer *i*, $x_{ik}$ denotes the activation of the *k*- th neuron in the previous layer, $w_{jk}^{\left(i\right)}$ is the weight associated with the connection between neurons *k* and *j*, and $b_{j}^{\left(i\right)}$ is the bias applied to neuron *j* in layer $L_{i+1}$.

In the specific case of an autoencoder, this operation can be represented in a more compact, vectorized form, considering a single hidden layer in the encoder and a single output layer in the decoder. For an input vector $x\in R^{n}$, the encoder projects the data into a latent space $h\in R^{m}$(m<n) as follows:
(2)$$h=f^{h}\left(W^{\left(1\right)}x+b^{\left(1\right)}\right)$$
where $W^{\left(1\right)}$ is the weight matrix between the input and hidden layers, $b^{\left(1\right)}$ is the bias vector, and $f^{h}$ is the activation function used in the encoder. In the decoding step, the latent vector *h* is used to reconstruct the input, resulting in an output $\hat{x}$ given by
(3)$$\hat{x}=f^{o}\left(W^{\left(2\right)}h+b^{\left(2\right)}\right)$$
where $W^{\left(2\right)}$ e $b^{\left(2\right)}$ are the weight matrix and bias vector of the output layer, respectively, and $f^{o}$ is the decoder’s activation function. The model is trained to minimize the difference between *x* and $\hat{x}$.

As the autoencoder must reconstruct the input as accurately as possible during the training process, the parameters should be adjusted to minimize the reconstruction error between the original input data and the reconstructed output [18]. The reconstruction error can be quantified by the mean absolute error (MAE), mean square error (MSE), root mean square error (RMSE), or another similar metric. Analysis of the distribution of these metrics allows anomaly thresholds to be defined, with a substantial increase in the reconstruction error suggesting the presence of anomalies, indicating that the model is having difficulty reconstructing anomalous data with the same accuracy as normal data.

In the context of predictive maintenance, the autoencoder has proven particularly effective in the early detection of asset failures. Typically, the autoencoder is trained using monitoring data that represent the asset’s healthy operational behavior. When the trained autoencoder is later applied to operational data containing failures or degradation, it is unable to reconstruct the input as effectively, resulting in noticeable discrepancies between the input and output data. These discrepancies can then be used to identify anomalies in complex systems [16].

In addition, the autoencoder can be applied to diagnostic and prognostic tasks. When coupled with a classifier, it can help diagnose problems based on previously recognized patterns [19]. Similarly, by extrapolating the residuals or using them in supervised models, the autoencoder can be used to predict the future behavior of systems [19].

Although autoencoders share conceptual similarities with traditional dimensionality reduction techniques such as principal component analysis (PCA) and t-distributed stochastic neighbor embedding (t-SNE), their applicability to unsupervised fault detection presents distinct advantages. PCA, a linear method, is limited in its capacity to capture nonlinear relationships between variables, which are often critical in complex systems such as wind turbines. In contrast, autoencoders, being neural network-based models, offer nonlinear mappings and greater representational power, allowing them to learn more intricate latent structures. While t-SNE is effective for visualization purposes, particularly in two or three dimensions, it is non-parametric and non-invertible, making it unsuitable for reconstruction-based anomaly detection. Autoencoders, on the other hand, are trained to minimize reconstruction error, which directly supports the identification of deviations from normal operating patterns. These properties make autoencoders particularly suitable for condition monitoring tasks where subtle and nonlinear degradation behaviors need to be captured without supervision.

## 3. The Proposed Framework

The proposed approach relies on detecting fault degradation patterns using the extrapolation of the reconstruction error from a deep autoencoder trained exclusively with monitoring data from the machine’s normal operational condition. This study focuses on identifying and isolating potential divergences in monitoring measurements, which may indicate a fault. The proposed methodology is structured into five main stages, which are illustrated in Figure 1.

### 3.1. Step 1—Study of the System, Failure Modes, and Monitoring System

In this first stage, a comprehensive study of the operating principles of wind turbines and their main failure modes is carried out. The focus is on the complete architecture of the turbine, its internal interactions, and the joint impact of each component on overall performance, providing a solid basis for the subsequent stages.

The FMSA is used to map possible failure modes and their respective observable symptoms in the operational data. This process is conducted on the basis of a specialized literature review, the manufacturer’s technical documents, and a history of recorded failures, making it possible to identify which monitored variables are most likely to reflect changes in the system’s behavior during the degradation process. For each failure mode identified, typical symptoms that can be observed during its progression are associated, such as abnormal temperature variations in certain components, which make up the set of attributes that are candidates for input into the model.

Although FMSA offers methods for calculating prioritization indexes and incorporating detection scales, this work does not use these classifications. The emphasis is on establishing a link among physical symptoms, sensor signals, and degradation mechanisms to ensure that the attributes used as input in the detection models are technically justifiable and faithfully represent the manifestation of the failure process. This guarantees a set of key characteristics and parameters capable of distinguishing the identified failure modes and ensures that autoencoder training focuses on attributes based on real field evidence and not on simple statistical correlation or empirical heuristics.

This form of structuring by failure mode and symptom is supported by recent literature. According to [20], the clustering approach has proven to be highly effective in condition-based maintenance contexts, as it directs the selection of signals based on their observability in relation to each failure mode. The authors point out that an observability-oriented detection model outperforms generic models that do not consider the alignment between symptom and cause. Similarly, ref. [21] demonstrate that segmenting input data based on groups of variables related to specific failure modes rather than using a single set for all cases results in a significant increase in the sensitivity of multivariate methods such as PCA and autoencoders. Clustering by failure mode, according to the authors, helps to reduce statistical noise, improve interpretability, and allow the model to specialize in identifying degradation patterns that are distinct from each other.

Although FMSA can, in theory, map a wide range of failure modes, it is essential to recognize that not all of them can be incorporated into the predictive model, especially when there are no monitored attributes that reliably represent them. In many cases, the relevant symptoms are not recorded by the monitoring system or are of poor quality, which compromises their usefulness for modeling purposes. For this reason, only failure modes whose symptoms are measurable with adequate frequency and quality are effectively integrated into the set of attributes used for training.

However, one of the advantages of FMSA over purely empirical approaches is the possibility of considering failures not yet observed in the actual maintenance history, based on consolidated technical knowledge and evidence documented in the literature. This makes it possible to anticipate degradation behavior even for components that have not previously failed in the system under analysis. For example, even if a component such as the generator bearing has never failed in the field, it is known from studies and experience in similar systems that it may show a rise in temperature as a symptom of degradation. Information like this can be previously mapped in the FMSA and used to select relevant attributes to be implemented in the model.

FMSA, thus, acts as a bridge between expert knowledge and algorithm modeling, structuring input data based on reliable causal relationships. This alignment among fault, symptom, and sensor contributes to building more robust models, capable of identifying faults more sensitively, more accurately, and in advance.

### 3.2. Step 2—Acquisition of Monitoring Data
and Feature Engineering

At this stage, data acquisition is a determining factor in the robustness of the anomaly detection system. The dataset used in this study comes from sensors installed in a wind turbine in real operation, covering highly critical attributes such as rotation speeds, temperatures, and power levels, in order to capture a complete overview of operating conditions.

To train the algorithm, it is necessary to divide the database into two parts: one containing records representative of the turbine’s normal operating behavior and the other containing data associated with degradation conditions. This separation is necessary because the model is initially trained exclusively with data considered to be healthy, in order to learn the typical and recurring patterns of normal system operation. Based on this learning, the model develops an ability to accurately reconstruct normal inputs. Subsequently, when exposed to data containing signs of failure or degradation, the algorithm tends to present higher reconstruction errors, since these patterns were not present in the training set.

To characterize the normal operational state, historical records from failure-free periods or maintenance expert insights are utilized. In certain cases, reference data may also include readings collected immediately after repairs or during a predefined time interval preceding failure occurrences. Although it is not feasible to establish an absolute threshold for distinguishing between normal and abnormal conditions in real-world operational data, this strategy seeks to capture gradual deviations that diverge from standard functioning patterns.

Data quality is another fundamental aspect of this phase. It is imperative that records classified under “normal operation” do not contain significant gaps or systematic errors that could compromise the model training process. Therefore, preliminary data audits must be performed to correct inconsistencies and ensure the reliability of sensor readings. Only after this verification has been completed does the preprocessing phase begin, followed by the implementation of the fault detection model.

### 3.3. Step 3—Development of the Fault Detection Tool

With a solid foundation established in the previous stage, the third stage involves the development of autoencoder models. These models are designed to detect faults in the wind turbine system, based on each specific fault mode, taking advantage of both the detailed study of the system and the monitoring data. Figure 2 illustrates the substeps corresponding to Stage 3 of the research.

#### 3.3.1. Defining the Strategy for Using the Monitoring Data

As highlighted by [22], the available dataset allows the development of a number of algorithmic strategies for fault detection, each with its own particularities.

Strategy 1—Specific model for each wind turbine: This approach involves creating an individual anomaly detection model for each wind turbine. This strategy can offer good accuracy as the model is adapted to a specific system, taking into account the particularities of each turbine. On the other hand, there is a higher computational cost and effort involved in maintaining multiple models;Strategy 2—General model for all the turbines in a wind farm: In this case, a single anomaly detection model is created for all the wind turbines in a specific wind farm. The advantage of this approach is that machine learning algorithms can be trained on a more comprehensive dataset collected from multiple turbines, resulting in a more informative training set that can lead to a good detection model, at the cost of potential loss of accuracy due to not considering individual specificities of each wind turbine;Strategy 3—Specific model for each failure mode: This strategy focuses on creating specific predictive models for each failure mode, training the algorithms on data targeted to each specific failure. This requires the creation of an individual training set for each failure mode, grouping together similar failure data from all turbines, as well as including a set of data without failures.

In this work, an adaptation of Strategy 3 for the autoencoder development was selected, taking into account both the specificity of the failure modes and the particularities of each turbine. Based on the mapping of possible failure modes and their respective symptoms through FMSA and the analysis of monitoring data, customized autoencoders were developed for each failure mode, and for each turbine in particular. By training specific autoencoders, each focused on a particular failure mode, it becomes possible not only to reduce computational costs and processing time but also to increase the efficiency of the system, allowing for more accurate detection of the failure process. This approach also provides greater flexibility in adapting the models to operational variations and the environmental characteristics of each wind turbine. In addition, individualized training of the autoencoders provides better interpretability of the results, as the reconstruction error patterns can be traced directly to a specific failure mode and turbine. The system chosen for this strategy is detailed in Figure 3.

Some events are external to the operation of the wind turbines and, although they can affect the subsystem, they are not included in the analysis, such as stoppages caused by weather conditions or human interference. This delimitation aims to focus on failures endogenous to the plant, allowing a more in-depth understanding of the system’s vulnerabilities and critical points.

#### 3.3.2. Defining the Autoencoder Architecture

The architecture of the autoencoder (i.e., the arrangement of layers and dimensions of the neurons) must be carefully designed to balance complexity and generalization capacity. Various architectures, such as multi-layer perceptron (MLP), convolutional neural networks (CNN) or long short-term memory networks (LSTM), can be used depending on the specific requirements and the nature of the data.

In general, the input layer corresponds to the number of features in each time window, representing all the sensory variables. In the encoder, a dense layer arrangement is adopted whose number of neurons decreases progressively (e.g., 32-16-8). This architecture allows the model to learn increasingly compact representations of the data.

At the center of the network, the latent space layer contains fewer neurons because they are the compressed representation of the data. This layer acts as a bottleneck, forcing the model to learn the most important features and discard noise and redundant information.

In the decoder, the architecture mirrors that of the encoder, progressively expanding (e.g., 8-16-32 neurons) and reconstructing the original data based on the latent representation.

The process also follows the usual steps of adjusting parameters and evaluating the model:Definition of the loss function: Generally, the mean squared error or the mean absolute error are adopted as reconstruction metrics.Choice of optimizer: Optimizers such as Adam, root mean square propagation (RMSProp) or stochastic gradient descent (SGD) can be used, depending on the behavior of the model and the desired performance.Setting hyperparameters: This involves selecting the number of layers, number of neurons in each layer, dropout rate, regularization techniques, activation functions (e.g., ReLU, sigmoid or tanh), and number of training epochs.

When adjusting hyperparameters, different combinations are explored to find the configuration with the lowest reconstruction error in the validation data. The search for these parameters can be performed either manually or automatically, using algorithms such as grid search, which is a method used to go through a matrix of defined values. Techniques such as random search or Bayesian optimization can also be used, which usually converge more quickly to regions of global minimum. At the end of this phase, the autoencoder architecture has been adapted to the characteristics of the problem and validated in terms of its reconstruction capacity.

#### 3.3.3. Developing Autoencoders

This subsection deals with the process of developing the networks. It begins by describing the pre-processing of the data, which is a fundamental stage in guaranteeing the quality and consistency of the information supplied to the model. This is followed by the implementation and training of the autoencoders. This workflow allows the correct learning of normal operating patterns and enables the identification of possible faults or degradations in turbine performance.

Data pre-processing is a fundamental stage in preparing raw data for analysis. This process involves various activities aimed at correcting inconsistencies, removing noise, and standardizing the variables. This initial care has a direct impact on faster convergence during training and on the model’s generalization power, allowing for more reliable detection of any abnormal operating events.

Initially, the data is cleaned, which includes checking for and dealing with missing values. Where feasible, imputation techniques such as interpolation are applied to fill in measurement gaps over time and maintain signal continuity. If the missing data is too numerous or inconsistent, it may be necessary to discard certain samples so as not to compromise the learning of the model.

Next, outliers filtering is performed on the data from the normal operating state, in order to identify and remove values that deviate significantly from the turbine’s normal behavior. These values may correspond to measurement errors or very specific operating conditions that do not represent routine operation. Statistical methods such as standard deviation and interquartile range (IQR) are used for this purpose. The IQR corresponds to the difference between the third (Q3) and first quartile (Q1) of the data distribution, resulting in the lower and upper limits.

Any data point that falls below the lower limit is considered an outlier. These values are significantly lower than those of the majority of the data set and are potential candidates for removal or further investigation. On the other hand, any data point that exceeds the upper limit is also considered an outlier. These values are much higher than those of the majority of the dataset and require special attention.

Another fundamental step in data pre-processing is standardization, in which the values of each attribute are updated so that the resulting set has an average value equal to zero and a standard deviation of one. This standardization makes it possible to compare quantities with different scales and contributes to numerical stability and accelerated convergence of the model during training. This prevents variables with higher magnitudes from dominating learning.

Mathematically, standardization using the Z-score technique consists of subtracting the mean (μ) and dividing by the standard deviation (σ).

The autoencoder must be trained exclusively with normal operating data, so that the model learns the typical operating patterns of the wind turbine. During training, the autoencoder adjusts its internal weights to reduce the discrepancy between the inputs and the reconstructed outputs, promoting the internalization of the regularities present in the data.

At the end of the training periods, the model is submitted to a validation process in order to assess its generalization capacity and its effectiveness in detecting degradation. To do this, a separate training dataset known as the hold-out set is used, which can include both records of normal operation not seen during training and data representative of degradation conditions, to assess the model’s ability to detect deviations.

During this stage, performance indicators are monitored, such as the validation loss function and reconstruction curves over time, which help to verify whether the model is overfitting or underfitting. These indicators serve as a basis for determining whether the model has reached a satisfactory configuration or whether additional adjustments to the architecture or hyperparameters are required.

The evaluation of the autoencoder’s performance is conducted on the basis of test data, focusing on the model’s ability to reconstruct representative samples of normal operating conditions and to identify degradation patterns or anomalous conditions. Initially, the accuracy with which the model reproduces the input data is analyzed, which reflects the autoencoder’s effectiveness in capturing the intrinsic patterns of normal data. At the same time, the sensitivity of the model in detecting atypical behaviour is assessed, characterized by significant increases in the reconstruction error when subjected to operating conditions that diverge from the learned patterns.

Specific metrics are used to measure the quality of the reconstruction and the efficiency of the model in detecting anomalies, in particular, the MSE. This metric is widely used to quantify the difference between the signal sequences reconstructed by the autoencoder and the values observed in the real data. A substantial increase in MSE is a reliable indicator of anomalies, as it reflects the model’s difficulty in reconstructing data that deviates from previously learned patterns.

### 3.4. Step 4—Applying in SCADA Monitoring Data

The application of the developed tool was carried out in two complementary stages. Initially, the model was tested with simulated data, containing representative records of different operating conditions and an induced failure mode. This first phase made it possible to validate the model’s ability to reconstruct normal behavior and detect anomaly patterns in a controlled environment, in which failure is previously known and labeled. From these data, indicators were generated that demonstrate the model’s potential to support predictive maintenance strategies.

The tool was then applied to a set of real operational data from SCADA systems, covering the operational variables of wind turbines in operation. These data include historical records of continuous operation and documented failure events, allowing an analysis of a controlled environment that is closer to the industrial environment. Figure 4 presents the sub-steps that make up Stage 4 of this paper.

#### 3.4.1. Setting the Threshold for Fault Detection

Once the model has been trained, the identification of anomalies is based on the difference between the input values and the values reconstructed by the autoencoder. To formalize this process, a threshold is set for the reconstruction error which, when consistently exceeded, indicates the presence of anomalous behaviour.

The anomaly detection threshold was set based on a non-parametric approach, using the value corresponding to the 99.7% percentile (percentage equivalent to the mean plus three times the standard deviation in a normal distribution) of the MSE obtained during the autoencoder training phase. This choice is justified by the lack of adherence of reconstruction errors to a normal distribution, which makes it impossible to apply parametric criteria based on mean and standard deviation. Let $\epsilon =\{\epsilon_{1},\epsilon_{2},...,\epsilon_{k}\}$ be the set of reconstruction errors, where each error is defined as(4)$$\epsilon_{i}=\left|\right|x_{i}-{\hat{x}}_{i}{\left|\right|}^{2}$$
with $x_{i}$ representing the input and ${\hat{x}}_{i}$ the respective reconstruction generated by the autoencoder. The threshold τ is set as(5)$$\tau =P_{99,7}\left(\epsilon \right)$$
where $P_{99.7}\left(\epsilon \right)$ corresponds to the value below which 99.7% of the reconstruction errors observed during normal operation lie. This value represents an upper tolerance limit, without the need to assume symmetry or any parametric form, normality, or underlying statistical distribution. The choice of percentile is based on coverage and safety criteria, seeking to restrict the signaling of anomalies to significant deviations from the behavior reconstructed by the model, regardless of the probabilistic structure of the data. Anomalies are detected by checking whether the reconstruction error of a new observation $\epsilon_{j}$ satisfies $\epsilon_{j}>\tau$. However, to avoid undue alarms caused by transient fluctuations or occasional noise, a temporal persistence rule was adopted. An event is classified as an anomaly only when *n* consecutive samples exceed the τ threshold.

#### 3.4.2. Defining the Persistence Rule for Fault Classification

After the data have been reconstructed by the autoencoder and the reconstruction error vector has been calculated, an additional verification routine is adopted to decide whether degradation is actually present. This approach seeks to deal with false positives arising from point variations or noise, ensuring greater reliability in the detection of anomalies.

Fault detection is based on a persistence criterion, which assesses whether reconstruction errors remain consistently above the established threshold. If a sequence of consecutive MSE values exceeds the threshold, an accumulative anomaly is characterized, indicating the start of degradation behaviour.

The persistence logic was implemented using a sliding window of fixed size *n*, applied to the reconstruction error vector. The value of *n* is adjustable and can be set based on the characteristics of the system or the desired level of sensitivity, preferably calibrated with expert support. Operation is as follows:**Persistence for activation of fault state (state 1):** The system monitors the reconstruction error vector over time, applying a sliding window with *n* consecutive readings. If all the values within this window are greater than or equal to a certain threshold, the system changes the state to failure (state 1), indicating the detection of a persistent anomalous condition. This criterion was established to ensure that small momentary deviations in the error data do not result in incorrect fault detection.**Persistence for return to normal state (state 0):** Similarly, reversion to the normal operating state (state 0) only occurs if *n* consecutive readings in the sliding window show reconstruction error values below the threshold. This criterion prevents a brief correction in the system’s behavior, which may be transient, from being interpreted as a complete return to the normal condition.

This persistence introduces an additional layer of robustness to fault detection, as it prevents the system from repeatedly switching between states in response to momentary fluctuations or noise in the data, thus ensuring that the transition between states only occurs in cases of consistent variations. This approach is in line with the need for stability in industrial systems, where temporary variations are common and where the early and consistent identification of faults can avoid unnecessary interventions and high operating costs.

For this work, *n* = 20 was adopted, i.e., when 20 points exceed the established threshold, the machine was classified as failing. This persistence was defined to reduce false positives caused by noise or transient weather phenomena, which can temporarily interfere with sensor readings, since the aim is to identify only progressive faults. These points are equivalent to 3 h and 20 min of machine operation. It should also be noted that the selection of this value is based on a global analysis of the system and the knowledge of experts in the field, as too few points could generate false positives, while too many would delay decision-making.

#### 3.4.3. Using the Tool in an Operational Context

The application of the tool in an operational context follows the flow presented in the Figure 5, detailing the stages of the monitoring and fault detection process. The process begins with the acquisition of monitoring data from sensors installed in the turbine. These data are processed by the autoencoder, which calculates the reconstruction error by comparing the input values with the values reconstructed by the model, previously trained under normal operating conditions.

The first analysis is to check whether the reconstruction error exceeds the previously defined threshold, which represents the expected range of variation for normal data. If the error remains below this threshold, the machine is considered to be in a normal operating state, reflecting stable and adequate system conditions. On the other hand, if the reconstruction error exceeds the threshold, an additional analysis stage begins, which assesses the persistence of this anomalous behavior.

Checking for persistence avoids generating false alarms due to temporary fluctuations in the data, such as noise or transient variations. Only when the reconstruction error remains consistently above the threshold over a predetermined period does the system classify the machine as being in a fault state.

### 3.5. Step 5—Performance Metrics Evaluation

The final stage of this research aims to discuss the applicability of the proposed model for automatic fault detection, by analyzing its performance on both simulated and real wind turbine operating data. After training the autoencoder, the models generated are experimentally validated using a separate dataset exclusively for testing.

When a labeled database is available, i.e., one in which normal operation records and fault events are previously identified, it becomes possible to use specific evaluation metrics in addition to the average reconstruction error. Metrics such as the true positive rate, which represents the proportion of faults correctly identified, and the false positive rate, which indicates the proportion of normal conditions incorrectly classified as anomalies, can be used for a more robust quantitative assessment. These metrics are calculated exclusively during the testing phase, from a reference vector containing known periods of degradation. Thus, initially, the model is then evaluated in a controlled environment, using simulated data with known labels, in which the periods of failure and degradation are explicitly defined.

Based on the results obtained from the simulation data, the model is then applied to a set of unlabeled real data from the SCADA system. In this scenario, the absence of explicit labels makes it impossible to use supervised metrics directly. Thus, the evaluation is based on anticipating the fault record present in the EDP’s operational log, using the temporal persistence rule to ensure that the fault classification is robust and not influenced by transient fluctuations in the reconstruction error.

In addition, the local variability present in the data is taken into account, especially in equipment subject to multiple operating modes, intermittent operating cycles or natural transitions between operating states, since fluctuations in reconstruction values can directly affect the network’s ability to detect anomalies. In these situations, the dynamic behavior of the machine can generate momentary fluctuations in the reconstruction error, which do not necessarily indicate faults. These fluctuations can compromise the sensitivity of the model and lead to misinterpretations caused by occasional fluctuations or noise in the data.

For this reason, the definition of the detection threshold and the size of the persistence window must take into account the operational characteristics of the monitored equipment, ideally with the help of experts. This phase is crucial to ensure a balance between sensitivity (the ability to identify real faults) and specificity (avoiding false positives).

## 4. Results

This chapter examines the applicability of the formulated method and presents the results of applying autoencoder techniques to a simulated database and a real wind farm operation database. Finally, it discusses limitations, further improvements, and future opportunities.

The incorporation of expert knowledge into the proposed methodology is formalized through the development of the FMSA, which serves as a cornerstone for fault-mode-driven variable selection and system understanding. This analytical framework was constructed with the active participation of three highly qualified experts, all holding doctoral degrees in engineering. Two of them are recognized specialists in reliability engineering, with extensive experience in methodologies such as FMEA, FMSA, and failure diagnostics across complex industrial systems. The third expert brings deep practical expertise in wind turbine operation, maintenance routines, and failure mechanisms, having worked directly with utility-scale wind energy systems. Their combined contributions ensured that FMSA was not merely a theoretical construct, but a rigorously validated framework grounded in field knowledge and reliability science.

### 4.1. Simulated Data

To support the initial validation of the proposed methodology, a simulated database was employed to assess the model’s effectiveness in detecting structural faults in wind turbines. The simulations were based on the 5MW reference wind turbine model developed by the National Renewable Energy Laboratory (NREL), which is widely adopted in aeroelastic and structural dynamics research. These synthetic data were generated under controlled fault conditions, allowing for precise control over failure scenarios and the availability of fault labels information that is not present in the real SCADA dataset provided by EDP. Such labeling enables the evaluation of detection performance through objective metrics, such as accuracy, sensitivity, and detection time. Additionally, the use of simulated scenarios facilitates consistent benchmarking and reproducibility in future comparative studies with other anomaly detection techniques. The simulation and post-processing procedures are thoroughly described in [23], including the fault injection mechanisms and modeling assumptions, which enhance the transparency and interpretability of the results.

The simulation modeling was conducted in the OpenFAST software, and the database was provided by the *Wind Engineering Laboratory (LEVE)* of the University of São Paulo. The dataset covers nine normal operating conditions, generated from the combination of three average wind speeds, 7 m/s (torque control region), 10.5 m/s (transition zone), and 14 m/s (above the nominal speed), with three levels of turbulence (12%, 14%, and 16%), according to the criteria of the [24] standard.

To represent the failure condition, the presence of a mass imbalance of 1% at the end of one of the blades was simulated, in accordance with the guidelines of the [24] standard, which consider failures induced by ice formation or loss of material due to atmospheric discharges. This level of asymmetry represents an incipient but realistic fault, capable of generating centrifugal forces and anomalous vibrational patterns, without compromising the turbine’s continuous operation.

The methodology for generating the simulated data and the details of the modeling are described in [23], which presents a structured approach for synthesizing conditional monitoring signals using OpenFAST. The study provides complete documentation of the turbine adopted, including geometric, structural, and dynamic properties, as well as the signal processing flow required to transform the raw data from the simulation into a structured database suitable for the application of machine learning and fault detection techniques.

The final database contains 11,232 simulated records of normal operation and 442 records of unbalanced turbine conditions. The selection of the model’s input attributes is based on the expected sensitivity of the signals to structural variations caused by unbalance, also provided by the work of [23].

An FMSA analysis was carried out on the simulated database, as illustrated in Table 2, from which the most relevant attributes for detecting the fault under study were selected. Next, the variables representative of the machine’s operating conditions were identified, with the aim of contextualizing the behaviour of the signals and ensuring that the analyses were consistent with the equipment’s operating regime.

For the anomaly detection algorithm, a fully connected autoencoder with symmetric architecture and five hidden layers was used. Training was conducted only on normal operation data. The autoencoder hyperparameters, such as the number of layers, units per layer, activation functions, and optimizer, were selected through a structured empirical process, rather than via exhaustive grid search or automated tuning algorithms. This decision is justified by the nature of the datasets used in this study, both simulated and real-world operational data, which are relatively low-dimensional and exhibit limited structural variability. Under such conditions, highly complex or over-parameterized architectures are not required to achieve adequate reconstruction performance. Given the goal of detecting deviations from normal operation in relatively stable signals, simpler architectures tend to generalize better, reduce overfitting risk, and allow for efficient training and deployment. The adopted architectures were refined iteratively based on validation loss trends and empirical evaluation of reconstruction error behavior, balancing model capacity and robustness. This approach aligns with findings from previous literature, where lightweight autoencoder configurations have been shown to perform effectively in condition monitoring tasks when the input space is well behaved and domain-specific preselection of features is applied. Moreover, the use of the FMSA as a prior knowledge layer significantly reduces input dimensionality and noise, further mitigating the need for large-scale hyperparameter optimization. A summary of the final configurations adopted is presented in Table 3.

The dataset was stratified into three subsets: 7862 samples (70%) were used for model training, 1685 samples (15%) for validation during the weight adjustment process, and 1685 samples (15%) for final performance evaluation in the testing phase. All features were standardized using the z-score method to ensure the normalization of attribute scales and to improve training convergence.

After training, the model achieved a mean squared error of 0.0007821 on the training set and 0.000797 on the validation set, indicating good generalization capability and absence of overfitting. The detection threshold, corresponding to the 99.7% percentile of the reconstruction errors, was computed, resulting in a threshold of τ=0.004094.

Figure 6 illustrates the reconstruction error for the test samples under normal operating conditions. It can be observed that the majority of the points remain consistently below the threshold. This behavior is expected, as the autoencoder was trained exclusively on data representing normal system operation, learning to accurately reconstruct the typical patterns of the system. The low magnitude of the errors confirms the model’s ability to adequately represent normal behavior.

Although most of the data exhibited low reconstruction errors, occasional values exceeding the threshold were observed, resulting in false positives. These deviations can be attributed to noise, natural operational fluctuations, or local data peculiarities that were not fully captured during training.

Figure 7 illustrates the temporal evolution of the reconstruction error generated by the autoencoder for the simulated fault scenario involving a 1% mass imbalance in one of the turbine blades. The blue points represent the mean squared error values computed for each observation, while the red dashed line indicates the predefined anomaly detection threshold. The magenta-highlighted points correspond to samples that remain below the detection threshold. The dotted and dashed vertical lines indicate transitions in operational conditions during the simulation: Thin dashed lines mark changes in wind turbulence levels, while the black double-dashed lines denote changes in average wind speed.

It is observed that, from the second transition in wind speed onwards, the reconstruction error increases significantly and persistently. This behavior demonstrates that the autoencoder, trained exclusively on data representative of normal operation, was able to capture deviations in the anomalous behavior pattern and the shift in operating conditions. The increase in error beyond the predefined threshold indicates that the model responded sensitively to the onset of degradation.

The autoencoder model was evaluated using confusion matrix analysis applied to the combined test and fault dataset, comprising a total of 2128 instances. Figure 8 presents the confusion matrix derived from the reconstruction errors of the autoencoder, summarizing the relationship between the predicted values and the actual class labels “Normal” and “Fault”.

The resulting confusion matrix shows that 1649 observations corresponding to the normal operating state exhibited reconstruction errors below the defined threshold and were, therefore, correctly identified as regular operating conditions. Similarly, 441 records associated with faulty conditions presented reconstruction errors above the threshold, correctly indicating the presence of a fault.

However, 38 misclassifications were observed. In 37 of these cases, normal operating data produced reconstruction errors above the threshold, leading to their incorrect identification as anomalous conditions (false positives). Only one faulty instance exhibited an error below the threshold and was not detected by the model, constituting a false negative.

The quantitative performance analysis of the model, based on metrics derived from the confusion matrix, demonstrates a high discriminative capability for fault detection, as shown in Table 4.

The overall accuracy achieved was 98.2%, reflecting the total percentage of correctly detected instances. A precision of 99.9% was observed for the “Normal" class and 92.3% for the “Fault” class, indicating that most samples identified as normal or faulty indeed correspond to their actual class labels. The recall rate, which expresses the proportion of correctly identified instances within each class, was 97.8% for the “Normal” class and 99.8% for the "Fault" class, highlighting the model’s effectiveness in detecting anomalous events.

Additionally, the F1-Score, defined as the harmonic mean between precision and recall, reached 98.9% for the “Normal” class and 95.9% for the “Fault” class, reinforcing the model’s balance between sensitivity and specificity. Both macro and weighted averages also support the robustness of the model, with all metrics exceeding 96%, indicating consistent performance even in the presence of class imbalance.

It is important to emphasize that, since this is an unsupervised approach based on autoencoders, the separation between “Normal” and “Fault” instances does not result from a conventional supervised classification process. Instead, anomaly detection is conducted by comparing the reconstruction error generated by the model against a predefined decision threshold. The effectiveness of the method is, therefore, directly linked to the autoencoder’s ability to accurately reconstruct normal operating patterns, such that significant deviations in reconstruction error can be interpreted as potential indicators of failure.

Considering the practical application in a realistic scenario, in which fault identification is not based solely on checking the reconstruction error at each observation, but rather on the persistence of deviations over time, the simulated data representing the fault condition were submitted to detection logic using the temporal persistence rule, which establishes that the transition from the normal operating state to the anomalous state is only confirmed when the reconstruction error exceeds the detection threshold in a sustained manner for a window of points, in this case, n=20 consecutive samples.

According to the reconstruction error of the fault data, it was observed that the first observation of the set was below the threshold, as shown in Figure 7; however, the function responsible for detection only classified the change of state to an anomalous condition from the 21st observation, when the persistence of the error was validated, as illustrated in Figure 9.

After applying the persistence window, the model’s final metrics were obtained, now adjusted to reflect only sustained state changes over time. These classifications were then compared with the actual labels of the instances, enabling a quantitative assessment of the model’s performance. Figure 10 shows the resulting confusion matrix after classifying the persistence rule, accompanied by the main evaluation metrics presented in Table 5.

The evaluation of the performance metrics after applying the persistence rule shows the robustness of the proposed approach. The overall accuracy achieved was 99.0%, with high F1-scores for both the “Normal” class with 99.4% and the “Failure” class with 97.6%, indicating a balance between precision and recall.

The recall of the “Normal” class was 100%, which means that all instances of regular operation were correctly recognized, without generating undue alarms. This result reinforces the importance of the persistence rule as an effective strategy for mitigating false positives, by requiring that deviations in the reconstruction error remain above the threshold for a minimum time interval before concluding that a fault has occurred. This requirement acts as a temporal filter, capable of ruling out one-off fluctuations or transient noise that could otherwise lead to erroneous classifications.

On the other hand, the recall of the “Failure” class was 95.2%, reflecting the existence of 21 observations which, although fully associated with the anomalous regime, were not immediately classified as a failure. This stems directly from the time window defined in the persistence rule, which requires the anomalous condition to persist for at least 20 consecutive samples for the change of state to be confirmed. Although this approach introduces an inherent detection delay, it contributes to system reliability by prioritizing signaling stability and avoiding hasty detections.

Regarding the quantification of how early the model can detect anomalies before actual failure, it is important to highlight a key limitation of the simulated dataset. As described in [23], the simulation framework does not model the complete progression of failures until the end of the asset’s life. Instead, it introduces three predefined levels of degradation severity, which allow the system to distinguish between healthy and degraded states, but not to estimate time-to-failure or the full degradation trajectory. Consequently, the current model’s output enables the identification of anomalous behavior indicative of early-stage degradation but does not allow for the precise quantification of advance warning time. Extending the simulation framework to include full-life failure progression would be necessary to assess detection lead time with greater precision.

Initially, the proposed methodology was validated using simulated fault data, which provided a controlled environment for analyzing the model’s response to well-defined anomalous patterns. Given the promising results obtained in the simulation, we moved on to a more challenging stage, representative of the real context, using historical data from EDP’s wind turbine SCADA system. The aim of this transition is to assess the model’s generalizability in the face of the complexities inherent in field operation, such as noise, operational variability, transient events, and unmodeled uncertainties. The results obtained from applying the model to real data are presented below.

### 4.2. EDP Database

The dataset used in this study comes from the Energias de Portugal (EDP) Open Data (https://www.edp.com/en/innovation/data, accessed on 12 July 2024), collected from an offshore wind farm located in the Gulf of Guinea, West Africa. This dataset comprises continuous monitoring and fault history records for five wind turbines, equipped with SCADA systems that monitor several key parameters of the turbines’ main components, as well as environmental measurements. The data were collected at regular 10 min intervals between 1 January 2016 and 31 December 2017, totaling 81 features.

The features include environmental parameters, such as wind speed and direction, and turbine operating conditions, such as component temperature and active power. Of these, 25 features are related to the temperature of the main turbine components, the notation and description of which are detailed in Table 6. It is important to note that some features, although associated with the same physical attribute, provide different statistics, such as average, minimum, maximum, and standard deviation, for the five turbines named T01, T06, T07, T09, and T11.

According to the ISO 16079 [25], the evolution of a mechanical failure can be detected initially by anomalies in vibration signals, often acquired by condition monitoring systems (CMS), followed by oil degradation, acoustic noise, and temperature, which is usually monitored by the SCADA system, until the problem is visually perceived. However, due to limitations of the dataset employed in this study, the available monitoring variables are restricted to SCADA measurements. As such, the present methodology focuses exclusively on thermal variables, to identify anomalies indicative of potential failure. This constraint inherently reduces the observability of early-stage degradation processes and may delay detection in comparison to methodologies that leverage higher-sensitivity signals such as vibration. Despite this limitation, prior studies such as [26] have demonstrated that temperature remains a relevant indicator in the detection of specific failure modes, particularly in components such as transformers and bearings where thermal imbalance is a direct consequence of mechanical or electrical degradation. Therefore, although the absence of CMS data imposes constraints on fault detectability and temporal resolution, the use of SCADA-based temperature monitoring still allows for the identification of deviations consistent with abnormal operational behavior, especially in the later stages of fault development.

The dataset includes historical records, also called a logbook, referring to 28 anomalies documented over two years of operation. These events have been categorized by affected component, covering critical systems such as the generator, generator bearing, multiplier box, transformer, and hydraulic group. The records, or logs, detail relevant information about each occurrence, such as the type of failure, its severity, and the approximate times at which the events were detected or reported.

According to [10], the anomaly file is filled in by the turbine operators, recording the moment a fault was detected, a component was affected, or a repair was carried out at a specific time. However, during the analysis of the data made available by EDP’s Open Data platform, an inconsistency was identified, since it does not include the SCADA data for Turbine 09 that was not available, despite the fact that the logbook shows several anomalies attributed to this unit. This absence compromises the possibility of correlating recorded events with the operating history of the turbine in question, limiting its use in quantitative analyses or in the validation of models. Table 7 presents a summary of the faults reported in the logbook.

The wind turbines are class 2 according to the [24] standard, have three blades with a diameter of 90 m and a hub height of 80 m, and are equipped with asynchronous generators connected to three-stage planetary gearboxes. The maximum rotor speed is 14.9 rpm, with a nominal power of 2 MW and a nominal wind speed of 12 m/s [27]. The technical data sheet is summarized in Table 8.

### 4.3. Failure Mode and Symptoms Analysis

The analysis was conducted taking into account data availability and quality restrictions, so that only failure modes actually observed in the history could be analyzed. This decision aimed to optimize the study for failures that had concrete information in the maintenance log, minimizing the risk of including hypothetical scenarios or those that were poorly supported by real data.

In this sense, each fault identified was correlated with possible symptoms that manifest themselves before or during its occurrence, such as temperature variations. This mapping showed which variables are most relevant for the early detection of each fault, serving as a basis for the careful selection of signals to be used in training the autoencoder.

As a result of this targeted approach, a set of key characteristics and parameters capable of distinguishing the failure modes identified from the available field data were established, consolidated in Table 9. This means that autoencoder training can focus on a representative dataset, prioritizing variables whose relationship with failures is backed up by concrete evidence.

In addition, the FMSA adaptation used allows efforts to be concentrated on components and subsystems whose criticality has been confirmed by real occurrences, avoiding overloading the models with speculative or inadequately documented failure scenarios. As a result, the analysis increases the reliability and coherence of the set of variables that will be used in training, ensuring that each failure mode and its symptoms are given due prominence in the modeling and anomaly detection process.

### 4.4. Data Extraction

As highlighted by [22], there is evidence that incipient failures can manifest themselves in monitoring data up to 60 days before their effective detection in this database. In the context of this work, this hypothesis was adopted to delimit the period of separation of data for those classified as normal, in order to avoid the inclusion of possible anomalies in the training set of the detection model, as illustrated in Figure 11.

As described by [10], the log is filled in by the machine operators, which makes it impossible to be certain as to the exact moment when the failure began or whether the date recorded corresponds to a turbine shutdown for maintenance, as it is possible to observe signs of failures on dates prior to the official date of the log.

In order to analyze the fault data, a period of approximately 40 days prior to logging was defined for each failure mode studied, as the aim of this work is to evaluate the ability of the tools developed to detect the recorded failure, avoiding the inclusion of possible previous failures that were not actually recorded.

### 4.5. Data Preparation

After selecting the most representative characteristics, the data were subjected to a preparation process to make them suitable for the subsequent modeling stages. First, the data were cleaned, eliminating records with missing values. Subsequently, the data were converted and restructured into a format compatible with the analysis algorithms, so that all the variables were in numerical format and organized in a matrix structure.

The occurrence of atypical values in the data, commonly known as “outliers”, can compromise the accuracy and reliability of the autoencoder, since these extreme values reflect measurement oscillations or unusual operating conditions that do not represent the standard behavior of the system. After separating the records considered “normal”, i.e., with the machine operating without faults, specific filtering was carried out to remove possible outliers. This ensured that the dataset used for training was effectively representative of the standard operating regime.

It is important to note that this filtering applies exclusively to data labeled as normal, since the purpose is to eradicate possible incorrect readings from the sensors or discrepant values that would not manifest themselves in genuinely normal operating circumstances. Data classified as anomalous or faulty are not subject to this removal.

The identification and filtering of extreme points in normal data was carried out using statistical methods based on the standard deviation and the interquartile range. According to established standards, any datapoint that falls below the lower limit is considered an outlier. These values are substantially lower than the majority of the dataset and are potential candidates for removal or further investigation. Conversely, any datapoint that exceeds the upper limit is also considered an outlier. These values are significantly higher in magnitude than the majority of the dataset, which may warrant particular attention.

After the cleaning and filtering stage, the data were standardized using the Z-score technique. For this purpose, a statistical standardizer was used, which calculates the mean and standard deviation of the attributes from a reference subset. In this study, the reference sample was made up exclusively of data labelled as “normal operation”, in order to prevent the presence of anomalies from influencing the calculation of the standardization metrics.

This processing ensures that all attributes contribute equally to the training of the autoencoder model. Without this standardization, variables with greater numerical amplitude could dominate the learning process, compromising the model’s ability to capture subtle patterns or correlations between attributes of different scales. The decision to calibrate the standardizer only with normal data maintains the model’s consistency with its operating principle: learning to reconstruct only typical operating patterns, from which significant deviations are interpreted as potential anomalies.

### 4.6. Development of Autoencoder for Fault Detection

After preprocessing, the autoencoder model was implemented and trained. The selected architecture was the fully connected multi-layer perceptron. Training was performed with the hyperparameters described in Table 10, over 150 epochs, ensuring model convergence.

The autoencoder was developed to compress the input data (encoding phase) and then reconstruct them (decoding phase), leading the model to learn the fundamental characteristics of the normal operating regime. The reconstruction error is monitored during the use of the model, making it possible to detect behavior that deviates significantly from these patterns, which makes it possible to characterize anomalies or incipient faults.

To this end, essential hyperparameters were adjusted, such as the number of layers, the loss function, the optimization technique, and the activation functions. The final configuration sought to balance the model’s ability to generalize with the complexity needed to capture the nuances of operational behavior.

The training was carried out exclusively with data from normal operation, randomly split into 70% for training, 15% for testing, and 15% for validation. The random selection aims to ensure representative samples in each subset, preserving the model’s ability to generalize appropriately.

It is important to note that the selected hyperparameters were kept unchanged in the models developed for faults in the transformer, the generator bearing, the hydraulic unit, and the gearbox, as shown in Table 10, in order to ensure standardization and facilitate comparison between results. It is worth noting that for these failure modes, the number of input characteristics was between four and six. This consistency in the dimensionality of the data motivated the adoption of the same network architecture for these cases. Except for the generator, an extra hidden layer was added due to the six input attributes. Thus, the final architecture for generator fault has five hidden layers, made up of 5, 3, 2, 3, and 5 neurons, respectively.

### 4.7. Transformer Fault Detection

#### 4.7.1. Turbine 01

The first fault analyzed in the history was that of the transformer in Turbine T01, caused by problems with its cooling fan. According to the failure log, the event occurred on 11 August 2017, at 1:14 p.m. The reported fault was directly associated with the malfunction of the fan responsible for maintaining the transformer’s operating temperature within safe limits.

To characterize this anomaly and create an effective detection model, features reflecting the turbine’s operating conditions and the component’s thermal behavior were selected. It is important to highlight that, while temperatures were used for their ability to capture symptoms related to the analyzed fault, two additional variables were selected to represent the turbine’s operational conditions: total energy production (*Grid_Prod*) and wind speed (*Amb_WindSpeed*). Including these features allows the model to learn seasonal variations present in the data without incorrectly classifying them as anomalies. Consequently, the autoencoder can recognize transformer temperature patterns under varying turbine operating conditions.

Before selecting training features, the data were filtered based on generated power, as the goal is for the model to learn the normal operational behavior of the wind turbine. Therefore, only data recorded when generated power was greater than zero were used for training. Considering all data, except failure records, as normal could induce undesirable model behavior. Rare events, such as technical stops, might mistakenly be classified as anomalies; however, the main focus remains the identification of potential faults rather than detecting such technical events.

After applying this filter, three features were selected for autoencoder training, generated power, wind speed, and temperature, from the transformer’s three phases, aiming specifically at fault identification. The model was trained using data from 2016 to May 2017, considered normal, excluding failure records to avoid including failed data.

After training the autoencoder, reconstruction errors were evaluated based on one main metric: MSE of 0.00361, calculated on the training data. This quantitative indicator reflects how well the model reproduces the original data from the latent representations, and is essential to assess its generalization capacity.

To detect anomalies, the reconstruction error based on the MSE metric was adopted as the main indicator of deviations from the expected behavior. Significantly high values indicate that the input was not well reconstructed by the model, suggesting the presence of patterns outside the normal operating regime. The distribution of MSE values obtained with normal operation data was analyzed in order to establish a decision threshold capable of adequately separating normal instances from potential failures. According to the methodology adopted, the threshold was defined based on the 99.7% percentile of the distribution of reconstruction errors, resulting in a value of τ=0.0167.

Figure 12 illustrates the behavior of the reconstruction error over time, along with the corresponding classification of the system’s operational state according to the persistence rule. The test interval, delimited from 12 June 2017 to 11 August 2017, includes the failure registered in the logbook. A clear upward trend in the reconstruction error can be observed during this period, reflecting the autoencoder’s reduced ability to accurately reconstruct the input data as the system degrades. In parallel, the binary classification line highlights the detection of the fault condition based on the persistence of elevated reconstruction errors over a predefined threshold. This mechanism ensures that transient fluctuations do not result in false positives, reinforcing the robustness of the fault detection strategy. The graph, thus, evidences that the model successfully detected the degradation process prior to the fault registration, enabling early diagnosis and the potential for preventive intervention.

Fault classification is not continuous, due to the presence of a transient region in which the data are gradually migrating from the normal operating regime to a degraded condition. In addition, the number of readings considered in the persistence window of the detection function directly influences the rate of correct fault identification, which reinforces the importance of adopting a customized approach that can be adapted to the characteristics of each specific application.

Therefore, the evaluation of the persistence-based detection mechanism must be conducted taking into account the particularities of the dataset used, such as the presence of operational noise, the variability of environmental conditions, and the sampling frequency. These factors directly impact the sensitivity of the classification.

#### 4.7.2. Turbine 07

Another thermal-related failure in the transformer was identified in Turbine 07, with two occurrences documented in the logbook on 10 July 2016 at 03:46 and 23 August 2016 at 02:21, respectively. To investigate this event, the autoencoder model was trained using data from the full year of 2017, yielding a mean squared reconstruction error of 0.001058 and a detection threshold defined at τ=0.0114.

Figure 13 presents a comprehensive visualization of the reconstruction error over time, alongside the binary classification of the turbine’s operational state after applying the persistence rule. The plot reveals that the reconstruction error frequently exceeds the established threshold throughout the monitoring period, with marked peaks that reflect significant deviations from the learned normal behavior. This persistent elevation in reconstruction error suggests that the transformer’s operating temperature remained at abnormal levels for extended durations, potentially indicating ongoing degradation.

Figure 13 also illustrates the output of the persistence-based classification, which mitigates the influence of transient spikes by confirming a fault state only after sustained deviation. This dual representation allows for a clearer distinction between sporadic anomalies and genuine degradations. Notably, the model signals the onset of abnormal behavior several days prior to the failure entries in the logbook, underscoring the model’s capacity for early fault detection. Such anticipatory insights are crucial for guiding timely maintenance actions and avoiding unplanned outages or equipment damage.

### 4.8. Gearbox Fault Detection

This section addresses the failures that occurred in the gearbox of two different turbines, characterized by damage to the gearbox pump and the bearing of the assembly. These problems resulted in an abnormal temperature rise, culminating in the need to stop the machines for repairs. To detect these failures, specific signals of oil temperature (*Gear_Oil_Temp*) and bearing temperature (*Gear_Bear_Temp*), as well as energy production data (*Grid_Prod*) and wind speed (*Amb_WindSpeed*), were used.

#### 4.8.1. Turbine 01

A failure associated with the gearbox system was recorded in Turbine 01 on 2016-07-18 at 02:10. The origin of the fault was identified as damage to the gearbox pump, a critical component responsible for maintaining oil circulation and the proper lubrication of internal parts. The malfunction compromised the thermal stability of the system, resulting in overheating, increased mechanical wear, and ultimately, the shutdown of the turbine.

Figure 14 presents the evolution of the reconstruction error produced by the autoencoder throughout the monitoring period, along with the corresponding binary classification of the turbine’s operating state after applying the persistence rule. The model was trained using data from the full year of 2017, achieving a mean squared error of 0.000596, a root mean squared error of 0.024426, and a detection threshold of τ=0.0055.

#### 4.8.2. Turbine 06

Another fault analyzed occurred in the gearbox of Turbine 06 on 17 October 2017 at 08:38, caused by bearing damage. This issue intensified, requiring the turbine to be shut down for corrective maintenance.

Figure 15 displays the temporal behavior of the autoencoder’s reconstruction error alongside the corresponding operational state classification obtained through the persistence verification strategy. The model was trained using data from the entire year of 2016, yielding a mean squared error of 0.000507 and a detection threshold of τ=0.0058.

The reconstruction error curve reveals several moments in which the values exceed the defined threshold, indicating deviations from the learned normal behavior—consistent with the mechanical instability caused by bearing degradation. Below this curve, the classification output marks the transitions between normal and faulty states, determined by the persistence rule. This rule ensures that an anomaly is only flagged when a sequence of consecutive points remains above the threshold for a predefined duration, thereby filtering out transient noise and spurious peaks.

### 4.9. Gnerator Fault Detection

This section addresses generator failures in two different turbines, T06 and T07. In T06, there were five records throughout 2016:11 July, at 7:48 p.m., the generator needed to be replaced;24 July, at 5:01 p.m., a temperature sensor failure occurred;4 September, at 8:08 a.m., a high temperature error was detected;2 October, at 5:08 p.m., the generator’s cooling system and temperature sensors were replaced;27 October, at 4:26 p.m., the generator was replaced again;

For T07, damage to the generator was detected on 21 August 2017, at 2:47 p.m., resulting in an abnormal increase in temperature and the consequent shutdown of the machine for repairs. To detect these faults, specific temperature signals from the three phases of the generator were used, in addition to the temperature of the slip ring, called *Gen_SlipRing_Temp* in the SCADA system.

#### 4.9.1. Turbine 06

Inconsistencies in the SCADA dataset were identified, notably, a complete absence of sensor readings between 11 July 2016 and 30 July 2016, as evidenced in the graphical analysis. Figure 16 illustrates the evolution of the reconstruction error generated by the autoencoder, as well as the classification of the generator’s operational state over time based on the persistence rule. For this evaluation, the model was trained using data from the entire year of 2017, resulting in a mean squared error of 0.000969 and a detection threshold of τ=0.0074.

The reconstruction error signal highlights distinct deviations from normal operating conditions, particularly outside the gap in data acquisition. These deviations are reflected in the lower portion of the graph, where the binary classification output marks transitions between normal and faulty states. The persistence rule is applied to ensure that only sustained anomalies, rather than isolated spikes, trigger fault detection.

#### 4.9.2. Turbine 07

Figure 17 shows the reconstruction error produced by the autoencoder over time, along with the corresponding classification of operational states based on the persistence rule, for Turbine 07’s generator. The model was trained with data from the entire year of 2016, resulting in an MSE of 0.001315, RMSE of 0.036266, and a threshold of τ=0.0121.

### 4.10. Generator Bearing Fault Detection

The next failure analyzed is related to problems in the generator bearing of Turbine 07, recorded on 20 August 2017, at 06:08. This condition is considered especially critical, as it directly affects mechanical stability, generating excessive vibrations, substantially increasing the operating temperature of the assembly, and accelerating the wear of other components. If not promptly corrected, the failure may evolve into even more serious impairments in the turbine, resulting in high maintenance costs and prolonged periods of turbine inactivity. To train the network to detect this failure mode, the temperatures of the two generator bearings are used, in addition to the operational control features. Figure 18 illustrates the evolution of the autoencoder reconstruction error, used to detect faults in the generator bearing of Turbine 07. The model was trained with data between June 2016 and June 2017, achieving an MSE of 0.001256, an RMSE of 0.035447, and τ=0.0073.

The graph also shows, over time, the classification of operating states after the persistence verification stage, in which values above a certain threshold are categorized as failure. It is noted that, in three intervals prior to the official record, high temperatures occurred in the bearing, exceeding this threshold and leading to the failure signaling. Although such occurrences were not recorded, they constitute early indications of anomalies that, if investigated in a timely manner, could have avoided more serious damage.

The correlation between the bearing failure and the subsequent generator problem is evident and highly correlated, since only one day after the bearing failure, damage to the generator itself was reported, as detailed in the previous section. This chain of events highlights the relevance of continuous monitoring of operational parameters, since early intervention in the bearing could have prevented or minimized the extent of damage to the generator. Furthermore, the adoption of predictive approaches, through detection algorithms, increases system reliability and reduces the occurrence of unplanned shutdowns, optimizing resources and promoting greater operational safety.

Finally, these results highlight the need to more rigorously integrate data analysis with the practical experience of maintenance teams. Investing in systems capable of identifying failure patterns, cross-referencing information from performance indicators, allows us to anticipate problems and create assertive maintenance plans.

### 4.11. Hydraulic Group Fault Detection

Finally, the last analysis was performed on the hydraulic group of Turbine 11, with three records throughout the study period, all describing an “Error in the hydraulic group of the brake circuit”. The occurrences were recorded on 17 October 2016, at 5:44 p.m.; 26 April 2017, at 6:06 p.m.; and 12 September 2017, at 3:03 p.m. This failure is critical, since the brake circuit plays a fundamental role in the safety and operational control of the turbine. A defect in the hydraulic group can result in failures in the brake actuation, compromising the ability to stop or regulate speed properly, which increases the risk of mechanical damage and accidents.

The attributes *Hyd_Oil_Temp* and *Cont_VCP-Temp* were used to train the model to detect these failures. Hydraulic oil temperature directly affects fluid viscosity and, consequently, brake system performance, while control module temperature may indicate overloads or abnormal operating conditions.

Furthermore, Figure 19, Figure 20 and Figure 21 illustrate the reconstruction error of the autoencoders developed to detect faults in the brake hydraulic system of Turbine 11. The model was trained with a dataset covering selected periods between January and July 2016, January and February 2017, and June and July 2017. In the learning process, the network achieved an MSE of 0.002299 and τ=0.0129. The graphs also indicates, over time, how the operating states are classified, after the persistence verification step.

Unlike the networks developed for other systems, which identified faults both prior to and on the date recorded in the logbook, the model trained for the hydraulic brake system of Turbine 11 did not signal the anomaly precisely at the official event date. Nevertheless, anomalies were consistently detected in the days preceding the record, aligning with the model’s objective of early fault detection. This result is positive, as official records often occur after operational symptoms have already manifested, and early recognition of degradation is desirable. However, it would also be expected that on the day of the event, typically associated with machine shutdown or intervention, the system would still present abnormal conditions, which was not observed.

Several factors may explain this discrepancy, including the intermittent use of the brake, potential errors in the manual recording of event dates, and the limitations of the temperature sensor, which measures oil temperature indirectly and possibly at locations distant from the actual heating point. These findings highlight the importance of improving fault detection systems by relocating sensors closer to critical components, expanding monitored variables (e.g., vibration), and adopting more structured event recording procedures. Additionally, they emphasize the inherent challenges of diagnosing faults in systems with intermittent operation, underscoring the need for continuous refinement of both measurement and documentation strategies.

### 4.12. Final Considerations

Although the proposed autoencoder model bases its decisions on the overall reconstruction error, the interpretability regarding individual feature contributions is indirectly addressed through the use of FMSA. Rather than applying techniques to quantify feature importance, the methodology incorporates domain knowledge during the feature selection phase. Each variable used in the model is explicitly linked to observable symptoms of a specific failure mode, as identified through FMSA. Consequently, any increase in the reconstruction error inherently reflects deviations in variables that are diagnostically relevant.

Performance metrics such as confusion matrix, F1-score, precision, and recall were not computed for the real-world SCADA data due to the absence of labeled ground-truth information. Unlike the simulated dataset, where faults are explicitly injected and temporally annotated, real SCADA records lack detailed and reliable labels indicating the exact timing, location, or severity of failures. As a result, conventional classification metrics are not applicable. Instead, the evaluation on real data was based on qualitative and temporal analyses of reconstruction error behavior, cross-referenced with available logbook entries.

The Table 11 presents a compilation of the results obtained during the training and testing stages of the neural networks developed for fault detection in wind turbines.

The first column indicates the identification number of the turbine analyzed, while the second specifies the component in which the fault was recorded. The third column refers to the date on which the fault was first recorded in the logbook provided by EDP, which serves as a historical reference for validating the events detected.

The following columns show the model’s technical parameters, the fourth column shows the root mean square error value obtained when training the autoencoder with normal operating data, and the fifth shows the τ detection threshold, defined based on the 99.7% percentile of the training reconstruction error distribution. The sixth column shows the date corresponding to the first fault classification returned by the model after applying the persistence rule, representing the moment when the anomaly was effectively detected on a sustained basis. Finally, the last column shows the difference, in days, between the model’s detection date and the fault’s record in the logbook, showing the network’s ability to anticipate when the fault is detected.

The results obtained throughout this study demonstrate the efficiency of the approach based on autoencoders, trained exclusively with data representative of normal operation, complemented by a detection logic based on temporal persistence of degradation, which is a promising tool for implementing intelligent predictive maintenance systems in complex engineering systems, such as wind turbines.

In all the instances analyzed, the neural networks demonstrated the ability to identify anomalous patterns early, significantly in advance of the official failure record documented in EDP’s logbook. The anticipation interval varied between 32 and 60 days, offering an adequate time window for the maintenance team to adopt measures.

The detection thresholds were set based on the 99.7% percentile of the distribution of reconstruction errors observed during the training phase. This definition, combined with the individualized adjustment of the models based on the FMSA of each failure mode, ensured high levels of sensitivity, respecting the particularities of each subsystem analyzed and promoting more accurate and robust detection of anomalies throughout the turbine’s different operating regimes.

## 5. Conclusions

The application of advanced maintenance management techniques is essential for the evolution of modern industry. In this context, this study proposed a sophisticated, data-driven fault detection system autoencoder-based for fault detection in wind turbine operational data. The unsupervised method of the approach enables early anomaly identification without requiring labeled fault data.

Despite promising results, the study identified several challenges that must be addressed to fully leverage this technique, including the dependence on high-quality normal data, the criticality of input attribute selection, sensitivity to noise and outliers, and limitations related to thresholding and root-cause identification.

The quality and coherence of the logbook provided by EDP represented another critical issue: Inconsistencies, information gaps, and potentially erroneous records complicate result validation. For instance, the logs mention Turbine T09 failures, even though this turbine is not recorded in the SCADA system. The absence of detailed data, such as failure severity, precise timestamps, and the operational state of the turbine at the moment of failure, introduces uncertainty and hinders a reliable correlation between event records and the machine’s behavior.

As with any data-driven methodology, the proposed approach is inherently sensitive to data quality issues such as missing values, sensor calibration drift, and excessive smoothing in preprocessed signals. Missing values, if not handled properly, can distort the reconstruction error distribution and compromise the reliability of anomaly detection. Similarly, calibration drift over time can gradually shift the statistical properties of input variables, leading to false positives or delayed detection due to model environment mismatch. Moreover, over-smoothing during SCADA preprocessing may obscure subtle anomalies, reducing the model’s sensitivity to early degradation. Although basic preprocessing steps were applied to mitigate these effects, such as data cleaning and normalization, more advanced mitigation strategies could be explored in future work, including imputation techniques, drift detection algorithms, and adaptive retraining.

Nevertheless, it is important to note that the objective of this study is not to construct a generalized model applicable across all turbines and failure types in a unique model. The methodology adopts a modular structure, wherein dedicated autoencoder models are developed for each specific failure mode and turbine unit. As such, the lack of SCADA data for Turbine T09 merely precludes its inclusion in the current modeling scope, without compromising the methodological integrity or the results obtained for the other turbines with complete and consistent data records.

Integrating autoencoders with analytical techniques such as FMSA proved efficient in accurately localizing failure modes and their corresponding attributes stored in turbine data acquisition systems. By providing more precise analysis of affected signals and correlating various fault types, this integration supports improved maintenance decision-making. It not only enables anomaly detection but also provides details about fault types, facilitating more accurate corrective actions. Based on these results, it is possible to develop preventive maintenance plans aimed at ensuring operational reliability and efficiency, minimizing downtime, and extending component lifespan.

Regarding deployment feasibility, it is important to note that the proposed model is not intended for real-time execution. Given the slow and progressive nature of most degradation processes in wind turbine components, fault detection can be effectively performed in batch mode, with inference intervals on the order of hours or days. This temporal resolution is sufficient for early anomaly identification and does not compromise detection efficacy. As such, the computational requirements are modest, enabling the model to be executed periodically on centralized servers or edge devices with moderate processing capacity.

Threshold selection plays a critical role in the sensitivity and specificity of anomaly detection systems based on reconstruction error. There are multiple strategies available in the literature, such as fixed percentiles, statistical control limits, adaptive thresholds, and dynamic modeling approaches, each offering different trade-offs between detection accuracy and false alarm rates. In the context of this study, where real-world SCADA data lack labeled failure instances, selecting an “optimal” threshold becomes inherently challenging. Without ground-truth annotations, it is not possible to precisely quantify false positives or false negatives, and any thresholding strategy must be interpreted within this limitation.

Future work should focus on further system improvements, recommending advancements in several directions. Initial improvements could include automating analytical processes, such as employing clustering techniques to group similar anomalous behaviors without manual labeling, the development of user interfaces, and integration with remaining useful life (RUL) estimation methods. Additionally, a comparative evaluation between the proposed method and other unsupervised anomaly detection algorithms, such as isolation forest, the one-class SVM, and PCA is planned. This future analysis aims to assess the relative performance of each method in terms of sensitivity, specificity, and robustness under varying fault scenarios and noise levels. The experiments will be conducted under equivalent data and evaluation conditions to ensure fairness, and will also explore the influence of FMSA-guided feature selection across alternative algorithms, enabling a broader validation of the methodology’s generalizability and interpretability. Moreover, future research can explore more robust and potentially adaptive thresholding strategies, particularly under semi-supervised or expert-in-the-loop frameworks. These approaches are expected to improve the balance between detection sensitivity and false alarm mitigation in real-world operational contexts, especially when labeled failure data are scarce or unavailable. Adaptive mechanisms may also prove essential for accommodating evolving operational conditions and concept drift in long-term monitoring applications.

Therefore, it can be concluded that although the proposed method demonstrated satisfactory performance aligned with project objectives, it exists within an operational scenario filled with complexities, requiring continuous refinement, integration with other approaches, and post-processing adjustments. It is anticipated that this study can serve as a foundation for developing more robust monitoring and diagnostic systems, ensuring enhanced reliability, safety, and efficiency in wind power generation.

## Figures and Tables

**Figure 1 sensors-25-04499-f001:**
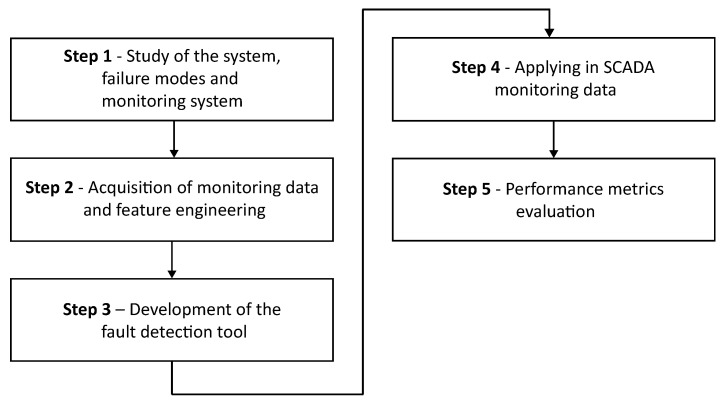
Flowchart describing the development of the proposed approach.

**Figure 2 sensors-25-04499-f002:**
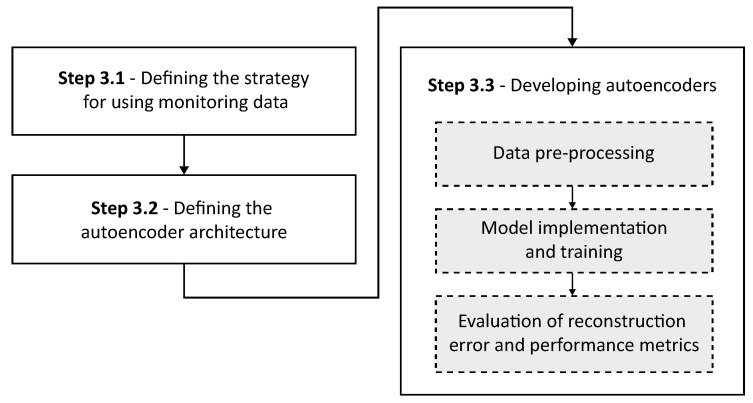
Visual representation of the sub-stages of step 3.

**Figure 3 sensors-25-04499-f003:**
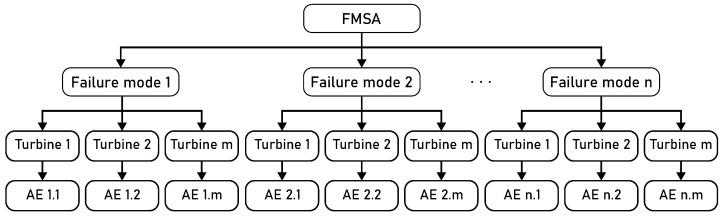
FMSA-guided development of turbine- and fault-specific autoencoder models.

**Figure 4 sensors-25-04499-f004:**
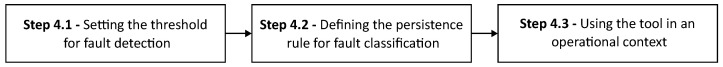
Stages of step 4.

**Figure 5 sensors-25-04499-f005:**
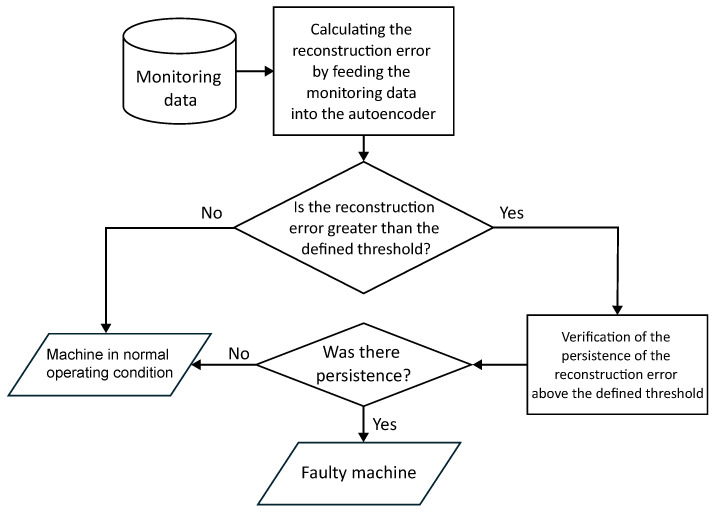
Flowchart of the application of the tool in an operational context.

**Figure 6 sensors-25-04499-f006:**
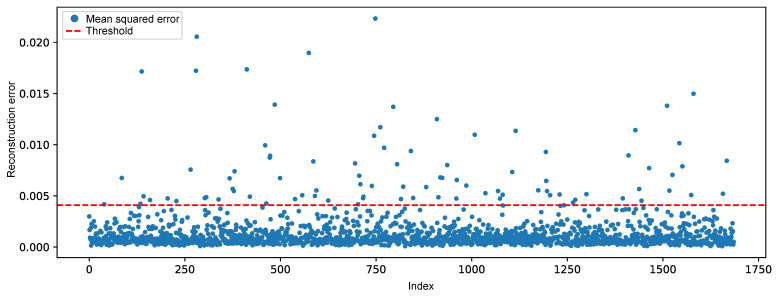
Reconstruction error over time of the simulated (normal) test data.

**Figure 7 sensors-25-04499-f007:**
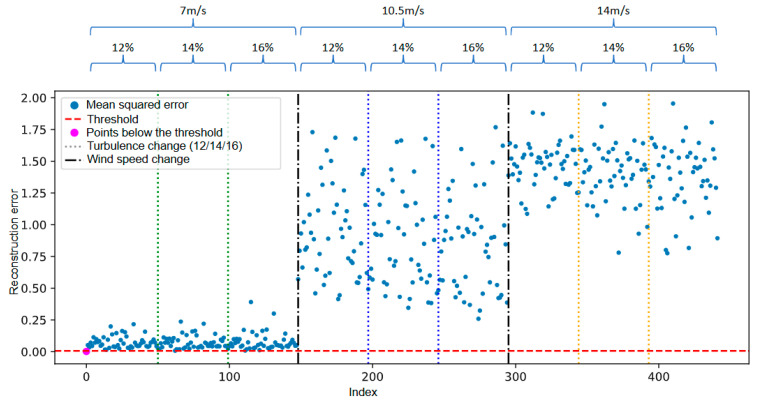
Reconstruction error over time of simulated fault data.

**Figure 8 sensors-25-04499-f008:**
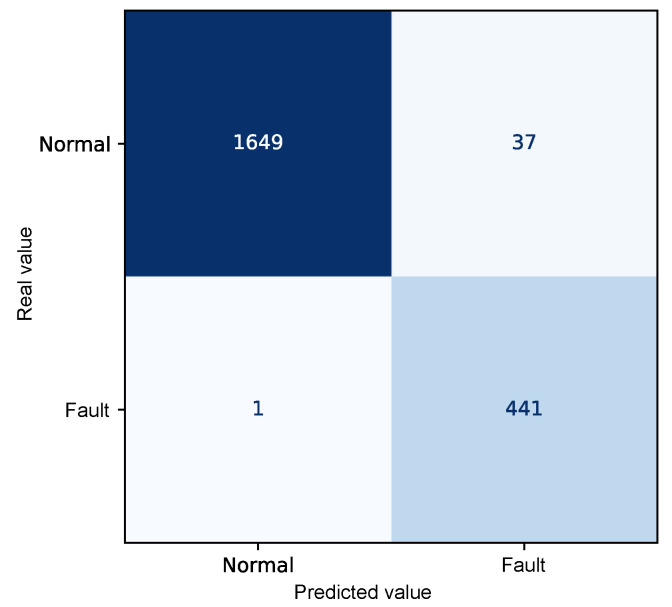
Confusion matrix relating the autoencoder’s reconstruction error to the detection threshold.

**Figure 9 sensors-25-04499-f009:**
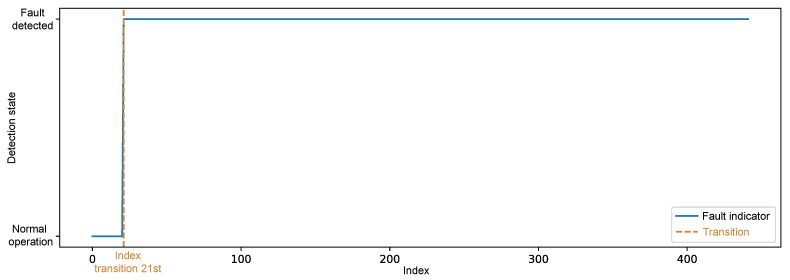
Detection state over time of simulated failure data.

**Figure 10 sensors-25-04499-f010:**
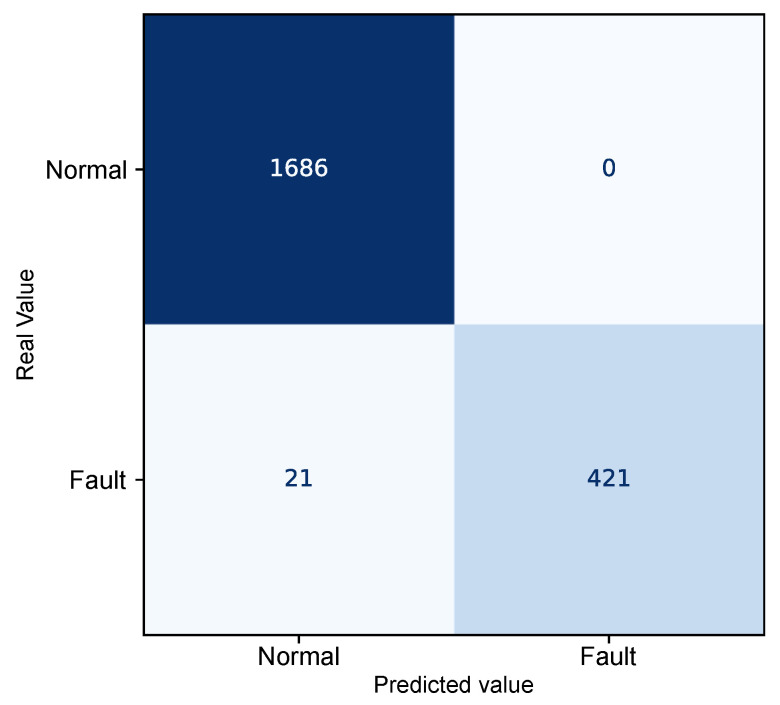
Confusion matrix after persistence window.

**Figure 11 sensors-25-04499-f011:**
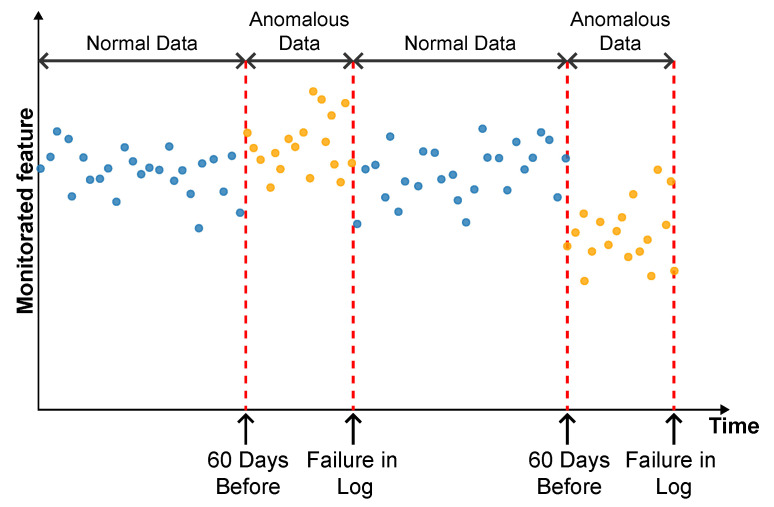
Extraction of normal data.

**Figure 12 sensors-25-04499-f012:**
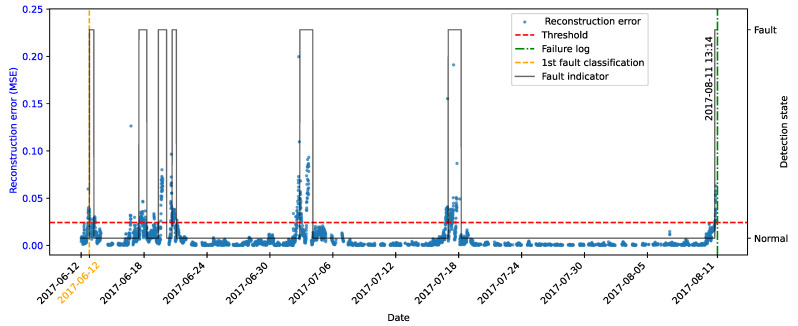
Observational reconstruction error and failure classification over time in Turbine Transformer 01.

**Figure 13 sensors-25-04499-f013:**
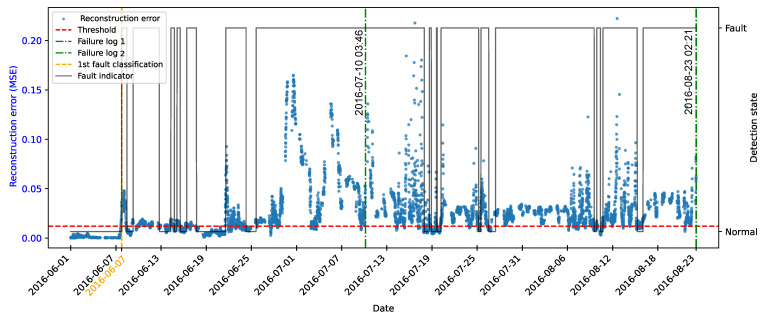
Observational reconstruction error and failure classification over time in Turbine Transformer 07.

**Figure 14 sensors-25-04499-f014:**
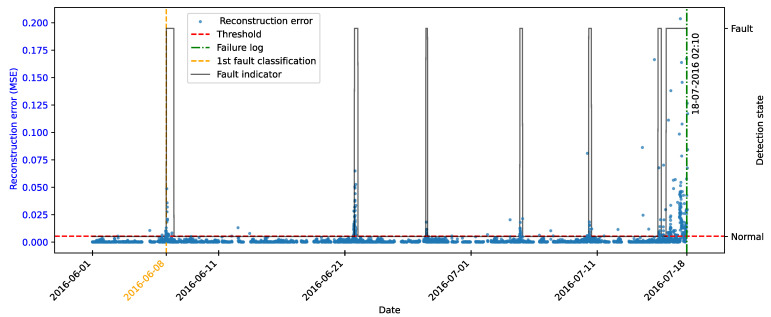
Observational reconstruction error and failure classification over time in Turbine Gearbox 01.

**Figure 15 sensors-25-04499-f015:**
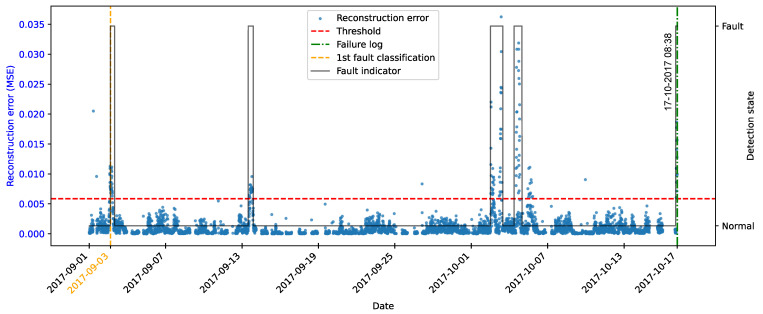
Observational reconstruction error and failure classification over time in Turbine Gearbox 06.

**Figure 16 sensors-25-04499-f016:**
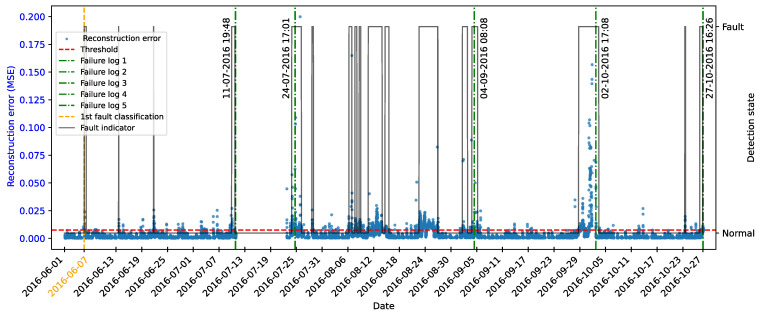
Observational reconstruction error and failure classification over time in turbine generator 06.

**Figure 17 sensors-25-04499-f017:**
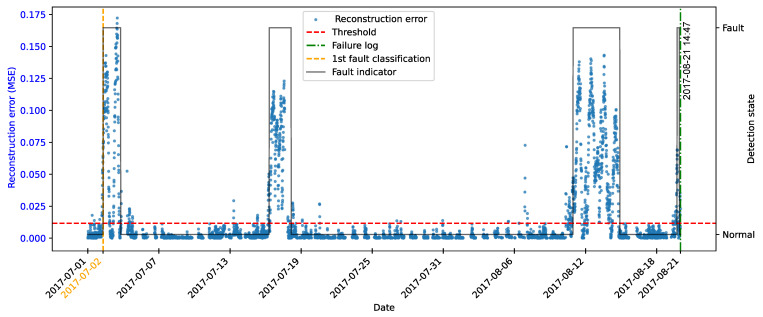
Observational reconstruction error and failure classification over time in Turbine Generator 07.

**Figure 18 sensors-25-04499-f018:**
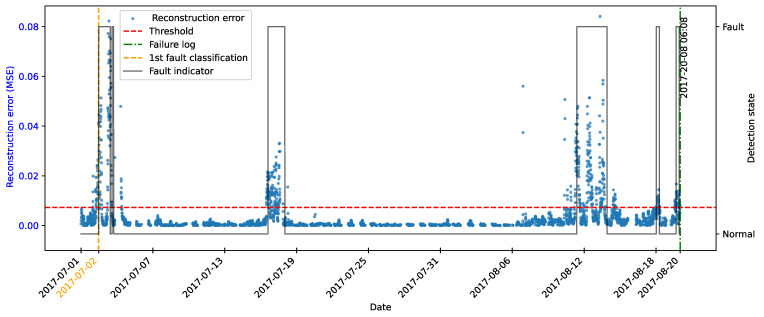
Observational reconstruction error and failure classification over time in Turbine Generator Bearing 07.

**Figure 19 sensors-25-04499-f019:**
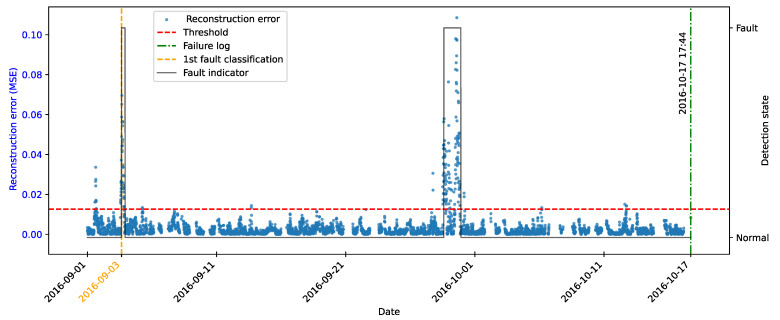
Observational reconstruction error and failure classification over time in Turbine 11 hydraulic group—October 2016.

**Figure 20 sensors-25-04499-f020:**
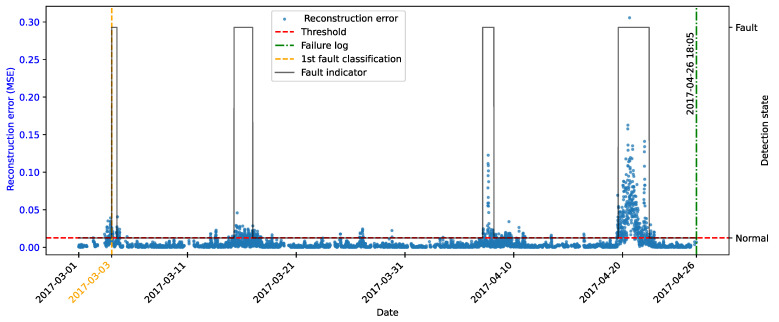
Observational reconstruction error and failure classification over time in Turbine 11 hydraulic group—April 2017.

**Figure 21 sensors-25-04499-f021:**
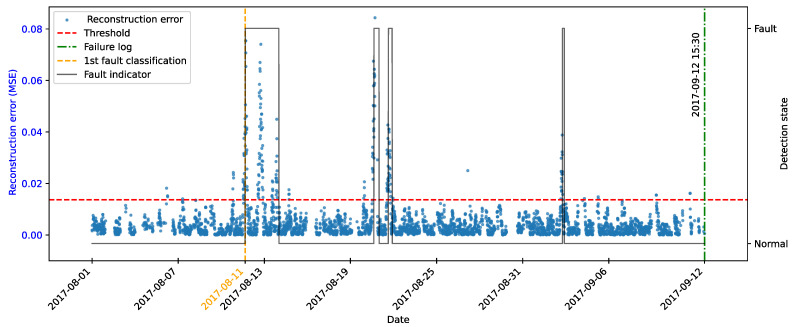
Observational reconstruction error and failure classification over time in Turbine 11 hydraulic group—September 2017.

**Table 1 sensors-25-04499-t001:** Comparison of recent studies on data-driven fault detection methods.

Study	System Under Study	Input Data Type	Detection Model	Feature Selection Method
Cui & Tjernberg [3]	Power transformers	SCADA (real)	AE + GRU	Not discussed (likely raw variables)
Radaideh et al. [4]	High voltage converter modulator system	Waveform signal (real)	LSTM, GRU, ConvLSTM AEs	No explicit selection; full waveform used
Tang et al. [2]	Wind turbine	SCADA (real)	Ensemble ML + adaptive threshold	Pearson, Spearman, variance filtering
Bindingsbø et al. [8]	Wind turbine generator bearing	SCADA (real)	XGBoost + SHAP	Domain-driven: selected 5 thermal/mechanical variables based on heat transfer physics
Chokr et al. [10]	Wind turbine	SCADA (real)	Bi-LSTM AE	Full SCADA window (no reduction discussed)
Miele et al. [11]	Wind turbine	SCADA (real)	GCN-AE + Mahalanobis	Sensor topology + time-series signals (graph-based); no explicit feature ranking
Huynh et al. [6]	Power grid	Phasor measurement unit signals (real)	Multi-layer AE	Layered PMU signals; no feature selection discussed
Nogueira et al. [12]	Wind turbine	SCADA (real)	MLP Autoencoder	Manual variable selection based on literature and correlation
**This work**	Wind turbine	SCADA (real) + OpenFAST (simulated)	AE guided by FMSA	Symptom-based selection guided by FMSA

**Table 2 sensors-25-04499-t002:** FMSA worksheet for the simulated fault.

**(a)**				
**Component**	**Function**	**Failure Mode**	**Failure Effect**	**Root Cause**
Rotor blade	Convert wind kinetic energy into rotational torque	Mass imbalance at the tip (1%)	Increased vibration	Material loss due to lightning strike
			Generation irregular centrifugal forces	Ice formation
(**b**)				
**Root Cause**	**Failure** **Symptoms**	**Detection** **Methods**	**Measurement** **Location/Signal**	**Monitoring**
Material loss due to lightning strike	Excessive vibrations in the nacelle	Vibration analysis	Accelerometers on the nacelle	Continuous monitoring via SCADA and CMS system
Ice formation	Noise	Visual inspection via drone	Blade tip	Continuous structural monitoring

**Table 3 sensors-25-04499-t003:** Autoencoder hyperparameters for simulated data.

Hyperparameter	Value
Number of input/output attributes	8
Number of hidden layers	5
Loss function	MSE
Optimizer	Adam
Validation fraction	15%
Kernel regularizer	L1 (0.001)
Kernel initializer	HeNormal
Activation function	LeakyReLU
Hidden layer sizes	(5, 3, 2, 3, 5)
Epochs	150

**Table 4 sensors-25-04499-t004:** Model evaluation of the reconstruction error detection model on simulated data from normal and faulty operation.

	Precision (%)	Recall (%)	F1-Score (%)	Support
**Normal**	99.9	97.8	98.9	1686
**Failure**	92.3	99.8	95.9	442
**Accuracy**		98.2	2128	
**Macro average**	96.1	98.8	97.4	2128
**Weighted average**	98.3	98.2	98.2	2128

**Table 5 sensors-25-04499-t005:** Classification performance evaluation after applying the persistence rule on simulated data of normal operation and failure.

	Precision (%)	Recall (%)	F1-Score (%)	Support
**Normal**	98.8	100	99.4	1686
**Failure**	100	95.2	97.6	442
**Accuracy**			99	2128
**Macro average**	99.4	97.6	98.5	2128
**Weighted average**	99	99	99	2128

**Table 6 sensors-25-04499-t006:** SCADA temperature features.

Variable	Local	Component
v1	Generator bearing 1	Generator
v2	Generator bearing 2 (drive-end)	Generator
v3	Generator stator winding phase 1	Generator
v4	Generator phase 2 stator winding	Generator
v5	Generator stator winding phase 3	Generator
v6	Hydraulic unit oil	Hydraulics
v7	Gearbox oil	Gearbox
v8	Generator bearing on high-speed shaft	Gearbox
v9	Nacelle	Nacelle
v10	High voltage transformer phase 1	Transformer
v11	High voltage transformer phase 2	Transformer
v12	High voltage transformer phase 3	Transformer
v13	Transmission network IGBT drive	Grid
v14	Nacelle top control	Controller
v15	Hub controller	Controller
v16	VCP board	Controller
v17	Split ring chamber	Controller
v18	Nose cone	Spinner
v19	Choke coil	Controller
v20	IGBT drive part of phase 1 inverter rotor	Grid
v21	IGBT drive part of inverter rotor phase 2	Grid
v22	IGBT drive part of inverter rotor phase 3	Grid
v23	VCP coolant	Controller
v24	Busbar	Grid
v25	Ambient temperature	Environment

**Table 7 sensors-25-04499-t007:** Example of summary of the turbine’s failure log.

Turbine	Component	Date	Description
T01	Transformer	2017-08-11T13:14	Damaged transformer fan
T06	Generator	2016-07-11T19:48	Generator replaced
T06	Generator	2016-07-24T17:01	Generator temperature sensor failure
T06	Generator	2016-09-04T08:08	Generator high temperature error
T06	Generator	2016-10-27T16:26	Generator replaced
T06	Generator	2016-10-02T17:08	Generator cooling system and temperature sensors replaced
T06	Gearbox	2017-10-17T08:38	Damaged gearbox bearing
T07	Generator Bearing	2016-04-30T12:40	High temperature on generator bearing (sensor replaced)
T07	Transformer	2016-07-10T03:46	Transformer high temperature
T07	Transformer	2016-08-23T02:21	High temperature on transformer. Cooling system repaired
T07	Generator Bearing	2017-08-20T06:08	Damaged generator bearings
T07	Generator	2017-08-21T14:47	Generator damaged
T11	Hydraulic Group	2016-10-17T17:44	Hydraulic unit error in brake circuit
T11	Hydraulic Group	2017-04-26T18:06	Hydraulic unit error in brake circuit
T11	Hydraulic Group	2017-09-12T15:30	Hydraulic unit error in brake circuit

**Table 8 sensors-25-04499-t008:** Technical description of each turbine.

Description	
Rated Power (kW)	2000
Cut-in Wind Speed (m/s)	4
Rated Wind Speed (m/s)	12
Cut-out Wind Speed (m/s)	25
Rotor Diameter (m)	90
Number of Blades	3
Maximum Rotor Speed (rpm)	14.9
Rotor Tip Speed (m/s)	70
Rotor Power Density 1 (W/m^2^)	314.3
Rotor Power Density 2 (m^2^/kW)	3.2
Type of Gearbox	Planetary
Gearbox Stages	3
Generator Type	Asynchronous
Maximum Generator Speed (rpm)	2016
Generator Voltage (V)	690
Grid Frequency (Hz)	50
Hub Height (m)	80

**Table 9 sensors-25-04499-t009:** Failure mode and symptoms analysis for wind turbine components.

Component	Function	Failure Mode	Failure Effect	Root Cause	Failure Symptoms	Detection Methods	Measurement Location	Monitoring
High Voltage Transformer (HV Transformer)	Raises the generated voltage to levels suitable for power transmission and distribution.	Cooling System Problems	Overheating	Ventilation system failure	Overheating	Temperature measurement	HV_trafo_phase1 HV_trafo_phase2 HV_trafo_phase3	Online
			Possible transformer failure	Obstructions in air passages	Deviation in temperature readings	Analysis of ventilation flow		
			Reduced efficiency	Low oil level in the transformer	Voltage drop	Inspection of ventilation system		
				Leaks in the cooling system		Monitoring of oil levels		
				Overload		Visual inspection		
				Temperature sensor failures		Analysis of temperature trends		
Hydraulic Group	Transfers hydraulic energy to the drive system.	Leaks or failure in the hydraulic pump	System inoperability	Damage to seals or connections	Abnormal noise or vibrations	Temperature measurement	Hyd_Oil_Temp Cont_VCP-Temp	Online
			Loss of hydraulic fluid	Wear or mechanical failure	Variable pressures	Vibration analysis		
				Fault in pump motor		Visual inspection		
				Blockage in intake system		Hydraulic pressure measurement		
Gearbox	Transmits and adjusts the speed of the generator.	Gear wear	System breakdown	Inadequate lubrication	Increased oil temperature	Temperature measurement	Gear_Oil_Temp Gear_Bear_Temp	Online
			Increased temperature	Coupling wear	Excessive noise or vibrations	Vibration analysis		
			Increased friction	Shaft misalignment				
			Failure in shaft alignment	Lubricant contamination				
				Gear tooth breakage				
Generator	Converts mechanical energy into electrical energy.	Problems in the electrical system	Interruption in generation	Overload or short circuit	Disconnection from the power grid	Temperature measurement	Gen_Phase1_Temp Gen_Phase2_Temp Gen_Phase3_Temp Gen_SlipRing_Temp	Online
			Power outages	Exposure to adverse environmental conditions	Increased noise or intermittent faults	Vibration analysis		
			Decreased efficiency	Voltage spikes	Sudden drop in power output	Current and voltage measurement		
				Corrosion on contacts	Increased temperature in components			
Generator Bearing	Supports the generator shaft.	Bearing failure	Damage to the generator shaft	Inadequate lubrication	Excessive noise	Thermography	Gen_Bear_Temp Gen_Bear2_Temp	Online
			Faults in shaft alignment	Oil contamination	Imbalance	Vibration analysis		
			Decreased efficiency		Failure to rotate	Visual inspection		

**Table 10 sensors-25-04499-t010:** Autoencoder hyperparameters.

Hyperparameters	Definition
Number of input features	5
Number of hidden layers	5
Loss function	MSE
Optimization technique	Adam
Kernel regularizer	L1 (0.001)
Kernel initializer	HeNormal
Activation function	LeakyReLU
Number of neurons in hidden layers	(3, 2, 3)
Epochs	150

**Table 11 sensors-25-04499-t011:** Summary of the case study results.

Turbine	Component	First Log (EDP)	Training MSE	Threshold Value τ	First Classification	Days in Advance
T01	Transformer	11 August 2017	0.0036	0.0167	12 June 2017	60
T07	Transformer	10 July 2016	0.0010	0.0114	7 June 2016	33
T01	Gearbox	18 July 2016	0.0005	0.0055	8 June 2016	40
T06	Gearbox	17 October 2017	0.0005	0.0058	3 September 2017	44
T06	Generator	11 July 2016	0.0009	0.0074	7 June 2016	34
T07	Generator	21 August 2017	0.0013	0.0121	2 July 2017	50
T07	Generator bearing	20 August 2017	0.0012	0.0073	2 July 2017	49
T11	Hydraulic group	17 October 2016	0.0022	0.0129	3 September 2016	44
T11	Hydraulic group	26 April 2017			3 March 2017	54
T11	Hydraulic group	12 September 2017			11 August 2017	32

## Data Availability

The original contributions presented in this study are included in the article. Further inquiries can be directed to the corresponding author.

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
