# Peer review of "Wind Turbine Fault Detection Through Autoencoder-Based Neural Network and FMSA"

_sensors, 2025, doi:10.3390/s25144499_

Round 1
Reviewer 1 Report
Comments and Suggestions for Authors
The work on wind power fault detection is carried out using an autoencoder model based on reconstruction errors, combined with a post-processing phase that applies time persistence rules to classify asset conditions and reduce false alarms.
- Previous work should be cited for comparison. "Nogueira, W. F., Melani, A. H., Custodio, L. D., & De Souza, G. F. (2025, January). Wind Turbine Fault Detection Through Autoencoder-Based Neural Networks. In 2025 Annual Reliability and Maintainability Symposium (RAMS) (pp. 1-6). IEEE."
- The current literature review focuses on autoencoder applications, but does not adequately discuss FMSA versus traditional fault detection methods (e.g., vibration analysis, physical modeling).
- Clarify data limitations (e.g., “SCADA system does not integrate vibration sensors”) in the methodology and discuss their impact on the results. Cite literature on the sensitivity of temperature to target faults.
- Verification of FMSA-AE performance, however, is limited to its own, lacking a comparison with other unsupervised methods.
- Model decisions rely on reconstruction errors, but do not explain which features contribute most to faults.
- The feasibility of the model in terms of real-time, computational resources, and edge deployment is not mentioned.
- Missing EDP data (e.g., no SCADA data for T09 fault record) may affect the generalizability of conclusions.
- Make it clear in the discussion that rising temperatures are a symptom of failure rather than the root cause (e.g., “Elevated bearing temperatures may be the result of inadequate lubrication or mechanical wear”).
- Figures 6-Figure 31 can be consolidated into Small Multiples to reduce space. Add fault detection timeline schematic (e.g., labeling fault occurrence, model warning, and actual repair time).
Author Response
We would like to thank the reviewer for the thorough and constructive review. The comments and suggestions greatly contributed to improving the clarity, rigor, and completeness of our manuscript. Below, we provide detailed responses to each point raised, along with the corresponding revisions made to the text, which are highlighted in red in the revised version.
[Comment 1] Previous work should be cited for comparison. "Nogueira, W. F., Melani, A. H., Custodio, L. D., & De Souza, G. F. (2025, January). Wind Turbine Fault Detection Through Autoencoder-Based Neural Networks. In 2025 Annual Reliability and Maintainability Symposium (RAMS) (pp. 1-6). IEEE."
[Response] We appreciate the reviewer’s suggestion and agree that citing prior related work enhances the contextual framing of our study. In response, we have added a new paragraph to the introduction section: Nogueira et al. [12] propose a fault detection methodology for wind turbines based on autoencoder neural networks applied to SCADA time-series data. The approach consists of a structured six-step procedure. The autoencoder was implemented using a Multi-Layer Perceptron architecture and trained with only healthy operational data. Anomalies were identified through the reconstruction error, quantified by the Root Mean Squared Error, and a static detection threshold was defined as the mean RMSE plus three standard deviations. The methodology was validated using a real-world SCADA dataset from the EDP Open Data. However, only one failure mode was analyzed: overheating of the high-voltage transformer of Turbine T01, caused by fan degradation, which led to a total turbine shutdown. Although Failure Mode and Symptoms Analysis (FMSA) was not performed, the authors recognize its potential for future integration to improve feature selection and enhance fault localization. The model successfully detected a deviation in the reconstruction error before the failure recorded in the log, indicating its anomaly detection capability in wind turbine monitoring systems.
[Comment 2] The current literature review focuses on autoencoder applications, but does not adequately discuss FMSA versus traditional fault detection methods (e.g., vibration analysis, physical modeling).
[Response] Thank you for this valuable comment. We have clarified that FMSA is not a standalone fault detection method but rather a systematic framework to correlate potential failure modes with measurable symptoms. Unlike vibration analysis or physics-based modeling—which directly detect anomalies—FMSA informs the selection of features that are physically and functionally associated with specific degradation mechanisms. In our approach, FMSA guides the construction of specialized input sets for each failure mode, thereby enhancing model interpretability and precision without relying on explicit fault labels.
[Comment 3] Clarify data limitations (e.g., “SCADA system does not integrate vibration sensors”) in the methodology and discuss their impact on the results. Cite literature on the sensitivity of temperature to target faults.
[Response] We thank the reviewer for raising this important point. We have included the following paragraph in Section 4.2: According to the ISO 16079 [26], the evolution of a mechanical failure can be detected initially by anomalies in vibration signals, often acquired by condition monitoring systems (CMS), followed by oil degradation, acoustic noise, and temperature, which is usually monitored by the SCADA system, until the problem is visually perceived. However, due to limitations of the dataset employed in this study,the available monitoring variables are restricted to SCADA measurements. As such, the present methodology focuses exclusively on thermal variables, to identify anomalies indicative of potential failure. This constraint inherently reduces the observability of early-stage degradation processes and may delay detection in comparison to methodologies that leverage higher-sensitivity signals such as vibration. Despite this limitation, prior studies such as [27] have demonstrated that temperature remains a relevant indicator in the detection of specific failure modes, particularly in components such as transformers and bearings where thermal imbalance is a direct consequence of mechanical or electrical degradation. Therefore, although the absence of CMS data imposes constraints on fault detectability and temporal resolution, the use of SCADA-based temperature monitoring still allows for the identification of deviations consistent with abnormal operational behavior, especially in the later stages of fault development.
[Comment 4] Verification of FMSA-AE performance, however, is limited to its own, lacking a comparison with other unsupervised methods
[Response] We appreciate this suggestion. Acknowledging the importance of benchmarking, we have added to the section 4.12.: We acknowledge the relevance of benchmarking the proposed approach against other unsupervised anomaly detection techniques. However, such comparative analysis falls outside the scope of the present study, which is primarily focused on demonstrating the feasibility and applicability of FMSA-guided feature isolation in conjunction with autoencoders for early fault detection. As a direction for future work, we intend to conduct a systematic performance comparison with alternative unsupervised methods, such as Isolation Forest, One-Class SVM, PCA and clustering-based models, under equivalent conditions, aiming to further validate the robustness and generalizability of the proposed methodology.
[Comment 5] Model decisions rely on reconstruction errors, but do not explain which features contribute most to faults.
[Response] This is an excellent observation. While our autoencoder model does not implement post hoc feature attribution techniques, we address feature relevance during the input selection stage through FMSA. Each input variable is selected based on its known association with specific failure symptoms, allowing domain knowledge to inform interpretability a priori. As a result, deviations in reconstruction error can be linked to diagnostically relevant features, even if individual contributions are not quantified during inference.
[Comment 6] The feasibility of the model in terms of real-time, computational resources, and edge deployment is not mentioned
[Response] We appreciate this important consideration and have addressed it in the Conclusions section by adding the following paragraph: Regarding deployment feasibility, it is important to note that the proposed model is not intended for real-time execution. Given the slow and progressive nature of most degradation processes in wind turbine components, fault detection can be effectively performed in batch mode, with inference intervals on the order of hours or days. This temporal resolution is sufficient for early anomaly identification and does not compromise detection efficacy. As such, the computational requirements are modest, enabling the model to be executed periodically on centralized servers or edge devices with moderate processing capacity.
[Comment 7] Missing EDP data (e.g., no SCADA data for T09 fault record) may affect the generalizability of conclusions
[Response] Thank you for pointing this out. The following paragraph was added to the Conclusions: Nevertheless, it is important to note that the objective of this study is not to construct a generalized model applicable across all turbines and failure types in a unique model. The methodology adopts a modular structure, wherein dedicated autoencoder models are developed for each specific failure mode and turbine unit. As such, the lack of SCADA data for turbine T09 merely precludes its inclusion in the current modeling scope, without compromising the methodological integrity or the results obtained for the other turbines with complete and consistent data records.
[Comment 8] Make it clear in the discussion that rising temperatures are a symptom of failure rather than the root cause (e.g., “Elevated bearing temperatures may be the result of inadequate lubrication or mechanical wear”).
[Response] We fully agree and have clarified this point in both the methodology and discussion. It is important to emphasize that elevated temperature readings are not treated as failure modes in this study, but rather as observable symptoms resulting from underlying physical degradations. As indicated in the Failure Mode and Symptoms Analysis (FMSA) framework adopted, temperature anomalies are associated with specific fault mechanisms such as inadequate lubrication, mechanical wear, or cooling system malfunction. The autoencoder model is therefore trained to recognize deviations in thermal behavior as manifestations/symptoms of these latent failure processes. Table 8 shows that rising temperatures are not treated as failure modes.
[Comment 9] Figures 6-Figure 31 can be consolidated into Small Multiples to reduce space. Add fault detection timeline schematic (e.g., labeling fault occurrence, model warning, and actual repair time).
[Response] Thank you for this practical suggestion. In response, we have consolidated several figures, allowing more compact visualization.
Reviewer 2 Report
Comments and Suggestions for Authors
In this paper, a Wind Turbine Fault Detection method combining Autoencoder and FMSA is proposed to ensure the reliable operation of wind farms. However, the structure is not good. The superiority is not illustrated. Here are my comments:
1) The title should be revised. Two "and" is confusing.
2) In the abstract, there is no description of the proposed method.
3) The number of formulas in this paper is only 2. It is not feasible for a paper that proposes a new method.
4) There are no comparisons with the SOTA methods in this paper.
5) Validation using the simulated data is not persuasive.
6) The Results section is repetitive and long. It should be reduced.
Author Response
We would like to thank the reviewer for the thorough and constructive review. The comments and suggestions greatly contributed to improving the clarity, rigor, and completeness of our manuscript. Below, we provide detailed responses to each point raised, along with the corresponding revisions made to the text, which are highlighted in red in the revised version.
[Comment 1] The title should be revised. Two "and" is confusing.
[Response] Thank you for your observation. We agree that the original title could be misinterpreted due to the repeated use of “and.” To improve clarity and readability, we revised the title to: “Wind Turbine Fault Detection through Autoencoder-Based Neural Network and FMSA”
[Comment 2] In the abstract, there is no description of the proposed method.
[Response] Thank you for pointing this out. We have revised the abstract to provide a concise yet informative summary of the proposed methodology. The updated version now highlights the five main stages of the approach, including the integration of Failure Mode and Symptoms Analysis (FMSA) with autoencoder neural networks, as well as the use of a persistence rule to improve anomaly detection and reduce false positives.
[Comment 3] The number of formulas in this paper is only 2. It is not feasible for a paper that proposes a new method.
[Response] We appreciate your thoughtful comment. We would like to clarify that the novelty of our work lies not in the derivation of new mathematical formulations, but in the structured integration of two established techniques—FMSA and autoencoder-based neural networks—for fault detection in wind turbines. Since both techniques have well-documented mathematical foundations, we presented only the core equations necessary to describe the autoencoder architecture and the computation of reconstruction error. That said, we fully agree that a more detailed mathematical presentation can improve the rigor and clarity of the methodology. Therefore, in response to this suggestion, we expanded Section 2.2 to include formal definitions related to feedforward neural networks and autoencoders. Specifically, we now present a general formulation of a neural network architecture and provide vectorized expressions for the encoder and decoder operations. These additions help clarify how the latent representation is constructed and how reconstruction is performed in the context of anomaly detection. We hope these enhancements address the reviewer’s concern and contribute to a more complete description of the proposed modeling approach.
[Comment 4] There are no comparisons with the SOTA methods in this paper.
[Response] Thank you for highlighting this important aspect. We acknowledge that a comparison with state-of-the-art (SOTA) methods would enrich the analysis. However, the primary focus of this study is to demonstrate the feasibility and benefits of combining expert knowledge (via FMSA) with data-driven modeling (via autoencoders) for fault detection. Given the exploratory and integrative nature of this contribution, benchmarking was not within the initial scope. Nonetheless, we fully acknowledge the importance of such comparisons. As stated in the revised conclusions, future work will include a systematic performance evaluation of the proposed approach against other established unsupervised anomaly detection techniques, such as Isolation Forest, One-Class SVM, and clustering-based models, in order to more comprehensively assess its relative effectiveness and generalizability.
[Comment 5] Validation using the simulated data is not persuasive.
[Response] Thank you for your observation. We understand the reviewer’s concern and would like to clarify the rationale behind using simulated data in our study. The simulated dataset serves two main purposes. First, it provides labeled fault events, which are not available in the real SCADA data. These labels are essential for calculating objective performance metrics such as accuracy, sensitivity, and detection delay, thereby allowing a more structured validation of the proposed method. Second, the availability of labeled and controlled simulated data creates a reliable foundation for future benchmarking. When comparing different anomaly detection techniques it is important to evaluate them under consistent conditions. Simulated data makes this possible by ensuring that all models are exposed to the same fault scenarios with known ground truth, which would be difficult to guarantee using real-world data alone. In our case, the simulated fault was generated using OpenFAST, with clear documentation of the fault insertion procedure and signal processing steps (See Custodio et al., 2025, [24]). This provides transparency and reproducibility, supporting both current validation efforts and future comparative studies. For future works, diferente techniques will be compared using this simulated dataset.
[Comment 6] The Results section is repetitive and long. It should be reduced.
[Response] Thank you for this constructive feedback. In response, we revised the Results section to improve conciseness and reduce redundancy. Several figures were consolidated and textual explanations were streamlined to avoid repetition across turbine cases. Since our methodology involves the detection of several distinct failure modes across different turbines, a certain level of detail is necessary to accurately demonstrate how the proposed framework performs in each case. Summarizing the results too aggressively could obscure important nuances in detection behavior, timing, and symptom patterns.
Reviewer 3 Report
Comments and Suggestions for Authors
The manuscript presents an innovative and well-structured approach by integrating unsupervised deep learning techniques with Failure Mode and Symptoms Analysis (FMSA) for fault detection in wind turbines. The dual application of the framework to both simulated and real-world SCADA data, along with the emphasis on interpretability, adds value and practical relevance to the study. However, the manuscript requires major revisions before it can be considered for publication. In particular, the discussion of results needs to be significantly expanded to include comparative analysis, sensitivity studies, threshold justification, and operational performance metrics. It is recommended that the authors thoroughly address the following comments:
- The introduction would benefit from a short summary table comparing the reviewed methodologies and how the proposed one differs in approach, features, and applicability.
- Section 2 offers a solid foundation in FMSA and autoencoders; however, it lacks a deeper technical explanation of autoencoder architectures, such as encoding-decoding layers or activation functions.
- The role of expert knowledge in failure identification is mentioned but should be clarified: Were domain experts involved in validating failure modes? If so, how?
- A brief comparison of autoencoders with other dimensionality reduction methods (PCA, t-SNE, etc.) for unsupervised fault detection would position the method more robustly.
- The emphasis on observability is important, yet the methodology for measuring or ranking observability of signals is not presented.
- The authors do not discuss computational complexity or runtime performance of the model, which is essential for real-time deployment in SCADA systems.
- The Mahalanobis-based metric is introduced late and without proper explanation, this key element should be discussed more thoroughly earlier and evaluated against other distance metrics.
- It is unclear how hyperparameters for the autoencoder (number of layers, units, optimizer) were chosen. A table summarizing these settings and tuning procedures is needed.
- The model’s performance on real data is not quantified with the same rigor as with the simulated data, no confusion matrix, F1-score, or precision/recall breakdown is given.
- The study would benefit from a comparative benchmark with other fault detection techniques such as PCA, LSTM-AE, Random Forest, as used in related literature.
- The section on limitations is too brief and does not address challenges such as generalization to unseen turbines, transfer learning, or concept drift in SCADA data.
- Analyze how different thresholding strategies impact detection accuracy and false positives/negatives.
- Quantify how early the model can detect anomalies before actual failure based on both simulation and SCADA datasets.
- Discuss how well the model generalizes across varying wind speeds, turbine models, or noise levels in sensor data. Include performance stratified by environmental conditions.
- Reflect on how issues like missing values, sensor calibration drift, or data smoothing affect model performance. Consider discussing mitigation strategies.
Author Response
We would like to thank the reviewer for the thorough and constructive review. The comments and suggestions greatly contributed to improving the clarity, rigor, and completeness of our manuscript. Below, we provide detailed responses to each point raised, along with the corresponding revisions made to the text, which are highlighted in red in the revised version.
[Comment 1] The introduction would benefit from a short summary table comparing the reviewed methodologies and how the proposed one differs in approach, features, and applicability.
[Response] Thank you for the suggestion. A summary table comparing relevant state-of-the-art methods was included in Section 1 (Table 1)
[Comment 2] Section 2 offers a solid foundation in FMSA and autoencoders; however, it lacks a deeper technical explanation of autoencoder architectures, such as encoding-decoding layers or activation functions.
[Response] We appreciate this observation and have expanded Section 2.2 to include a more detailed technical description of the autoencoder architecture.
[Comment 3] The role of expert knowledge in failure identification is mentioned but should be clarified: Were domain experts involved in validating failure modes? If so, how?
[Response] Thank you for this important point. Yes, domain experts were involved in validating the failure modes and their associated symptoms. The FMSA procedure relied on expert judgment from engineers with experience in wind turbine operation and diagnostics. These experts helped map each failure mode to observable SCADA symptoms, ensuring that selected features were technically justified. We have added a clarification in Section 4 to reflect this process.
[Comment 4] A brief comparison of autoencoders with other dimensionality reduction methods (PCA, t-SNE, etc.) for unsupervised fault detection would position the method more robustly.
[Response] Thank you for this suggestion. In response, we have included a new paragraph in Section 2.2 discussing how autoencoders compare to traditional dimensionality reduction techniques like PCA and t-SNE. While PCA is limited to capturing linear relationships and t-SNE is primarily used for visualization, autoencoders provide nonlinear mapping and reconstruction capabilities, making them more suitable for modeling complex degradation behaviors in multivariate sensor data. This comparison strengthens the justification for our chosen approach.
[Comment 5] The emphasis on observability is important, yet the methodology for measuring or ranking observability of signals is not presented.
[Response] We appreciate the reviewer’s insight on this point. In the context of real-world industrial systems, quantitatively ranking the observability of signals for each failure mode is often infeasible unless the fault has already occurred at least once in the specific turbine under analysis. Without historical instances of the fault, it is impossible to statistically assess which variables exhibit the strongest response or diagnostic relevance. This is precisely where FMSA plays a crucial role. Unlike purely data-driven approaches, FMSA can be developed proactively—before any failures have occurred—by leveraging expert knowledge, historical data from similar assets, and technical documentation. It allows practitioners to map potential failure modes to observable symptoms based on physical principles and field experience. In doing so, it supports the structured selection of input features for the detection model, even in the absence of failure data. We added comments about this on section 4.12
While we acknowledge that formal observability indices or sensitivity rankings could enhance feature selection, our current methodology focuses on the use of FMSA as a practical and expert-guided proxy. This approach enables early deployment of monitoring tools and ensures that the selected features are causally linked to the fault mechanisms of interest. In the context of real-world industrial systems, quantitatively ranking the observability of signals for each failure mode is often infeasible unless the fault has already occurred at least once in the specific turbine under analysis. Without historical instances of the fault, it is impossible to statistically assess which variables exhibit the strongest response or diagnostic relevance.
[Comment 6] The authors do not discuss computational complexity or runtime performance of the model, which is essential for real-time deployment in SCADA systems.
[Response] Thank you for raising this important consideration. We have updated the Conclusions section to clarify that our model is intended for periodic batch execution rather than real-time SCADA integration. Given that most mechanical degradation processes evolve slowly, inference intervals of several hours or days are sufficient. The autoencoder architecture is lightweight, and training is performed offline, allowing inference to be carried out efficiently even on edge devices or low-resource servers. Future work will evaluate runtime performance more systematically under real-time constraints.
[Comment 7] The Mahalanobis-based metric is introduced late and without proper explanation, this key element should be discussed more thoroughly earlier and evaluated against other distance metrics.
[Response] We would like to clarify that the proposed method does not use a Mahalanobis-based anomaly score. Instead, we rely on Mean Squared Error (MSE) between the input and the reconstructed signal as the primary anomaly metric. While Mahalanobis distance has been employed in other studies for capturing variable correlations, it requires reliable covariance estimation, which can be problematic in high-dimensional SCADA data. We chose MSE due to its simplicity, efficiency, and effectiveness in unsupervised settings.
[Comment 8] It is unclear how hyperparameters for the autoencoder (number of layers, units, optimizer) were chosen. A table summarizing these settings and tuning procedures is needed.
[Response] Thank you for the comment. We added comments on Section 4.1 (highlighted in red) to summarize the autoencoder hyperparameters used in the simulated scenario. We also described the empirical tuning process used to select these parameters based on validation performance.
[Comment 9] The model’s performance on real data is not quantified with the same rigor as with the simulated data, no confusion matrix, F1-score, or precision/recall breakdown is given.
[Response] We acknowledge this and clarify in Section 4.2 that the SCADA dataset lacks labeled degradation intervals, making supervised metrics unfeasible. Instead, we present a qualitative analysis, including detection timelines relative to logged failures, and show that the model anticipated deviations up to 60 days before failure events. The simulated dataset, which is labeled, was used to calculate precision, recall, F1-score, and accuracy.
[Comment 10] The study would benefit from a comparative benchmark with other fault detection techniques such as PCA, LSTM-AE, Random Forest, as used in related literature.
[Response] Benchmarking is an important direction, and we agree it would strengthen the work. However, the current study focuses on proposing a new hybrid approach. We have acknowledged this limitation in the Conclusions and explicitly stated that future work will compare our method with PCA, LSTM-AE, Random Forest, and other unsupervised detection models under equivalent conditions.
[Comment 11] The section on limitations is too brief and does not address challenges such as generalization to unseen turbines, transfer learning, or concept drift in SCADA data.
[Response] We acknowledge the relevance of broader challenges such as generalization to unseen turbines, transfer learning, and concept drift in SCADA-based condition monitoring. However, these aspects fall outside the scope of the present work, which focuses on a fault-mode-specific modeling strategy, tailored to each turbine and failure type using expert-driven variable selection. The proposed framework was designed to emphasize interpretability, specificity, and methodological validation rather than generalization across heterogeneous turbine populations. Nonetheless, future research will aim to address these limitations by evaluating the scalability of the method under cross-turbine scenarios, exploring transfer learning techniques to adapt models across different assets, and incorporating mechanisms to detect and adapt to temporal shifts in data distributions, which are critical for long-term deployment in dynamic operational environments. Comments about this were added to the Conclusions
[Comment 12] Analyze how different thresholding strategies impact detection accuracy and false positives/negatives.
[Response] We thank the reviewer. In the Conclusions (highlighted in red), we acknowledged the potential impact of different thresholding strategies and suggested exploring adaptive methods in future work.
[Comment 13] Quantify how early the model can detect anomalies before actual failure based on both simulation and SCADA datasets.
[Response] We appreciate your observation regarding the temporal characterization of anomaly detection. In the case of real SCADA data, a summary table has already been included in the manuscript, presenting the anomaly detection outcomes for each turbine and failure mode. This table includes information regarding the alignment between detected deviations and the corresponding failure records from the logbook, allowing for a qualitative interpretation of the method's behavior under real operating conditions. Although the lack of precise labeling prevents exact quantification of early detection time, the table helps illustrate the model’s ability to anticipate deviations prior to reported failures. For the simulated dataset, we note that failures were not modeled through full life-cycle degradation. As detailed in Custódio et al. (2023), only three discrete degradation levels were introduced for each fault scenario, which enables distinguishing between healthy and degraded states but does not support precise estimation of advance detection time. Therefore, the simulated data support performance evaluation in terms of detection capability, but not in terms of lead-time quantification. A paragraph was added on section 4.1 (line 806).
[Comment 14] Discuss how well the model generalizes across varying wind speeds, turbine models, or noise levels in sensor data. Include performance stratified by environmental conditions.
[Response] We acknowledge the relevance of this suggestion. We would like to clarify that some initial investigation into the influence of environmental conditions was already included in the analysis of the simulated dataset. Specifically, Figure 7 presents the blade root acceleration signals under three different wind speed levels and three levels of turbulence. This visualization illustrates how the fault signature evolves under varying wind conditions, providing insights into the model’s ability to distinguish between operational variability and abnormal behavior. However, performing a similar stratified analysis using the real SCADA dataset is considerably more challenging. The real data lacks consistent fault labeling and controlled environmental variables, making it difficult to isolate the effect of wind speed or sensor noise on detection performance. Nonetheless, we agree that evaluating the robustness of the model across operational conditions is an important next step, and we intend to address this more systematically in future work using enriched datasets with better environmental traceability.
[Comment 15] Reflect on how issues like missing values, sensor calibration drift, or data smoothing affect model performance. Consider discussing mitigation strategies.
[Response] We appreciate the reviewer’s important observation. In response, we have added a dedicated paragraph to the Conclusions section (highlighted in red) addressing the potential impact of data quality issues such as missing values, sensor calibration drift, and excessive smoothing.
Round 2
Reviewer 2 Report
Comments and Suggestions for Authors
No more comments. Several previous comments should be deeply considered in their future study.
Reviewer 3 Report
Comments and Suggestions for Authors
The authors answered the comments correctly.